# Structural basis of Ca$^{2+}$-dependent activation and lipid transport by a TMEM16 scramblase

Maria E Falzone[1], Jan Rheinberger[2], Byoung-Cheol Lee[2,3], Thasin Peyear[4], Linda Sasset[5], Ashleigh M Raczkowski[6], Edward T Eng[6], Annarita Di Lorenzo[5], Olaf S Andersen[4], Crina M Nimigean[1,2,4], Alessio Accardi[1,2,4]*

[1]Department of Biochemistry, Weill Cornell Medical College, New York, United States ; [2]Department of Anesthesiology, Weill Cornell Medical College, New York, United States; [3]Department of Structure and Function on Neural Network, Korea Brain Research Institute, Deagu, Republic of Korea; [4]Department of Physiology and Biophysics, Weill Cornell Medical College, New York, United States; [5]Department of Pathology, Weill Cornell Medical College, New York, United States; [6]Simons Electron Microscopy Center, New York Structural Biology Center, New York, United States

*For correspondence:
ala2022@med.cornell.edu

Competing interests: The authors declare that no competing interests exist.

**Abstract** The lipid distribution of plasma membranes of eukaryotic cells is asymmetric and phospholipid scramblases disrupt this asymmetry by mediating the rapid, nonselective transport of lipids down their concentration gradients. As a result, phosphatidylserine is exposed to the outer leaflet of membrane, an important step in extracellular signaling networks controlling processes such as apoptosis, blood coagulation, membrane fusion and repair. Several TMEM16 family members have been identified as Ca$^{2+}$-activated scramblases, but the mechanisms underlying their Ca$^{2+}$-dependent gating and their effects on the surrounding lipid bilayer remain poorly understood. Here, we describe three high-resolution cryo-electron microscopy structures of a fungal scramblase from *Aspergillus fumigatus*, afTMEM16, reconstituted in lipid nanodiscs. These structures reveal that Ca$^{2+}$-dependent activation of the scramblase entails global rearrangement of the transmembrane and cytosolic domains. These structures, together with functional experiments, suggest that activation of the protein thins the membrane near the transport pathway to facilitate rapid transbilayer lipid movement.
DOI: https://doi.org/10.7554/eLife.43229.001

## Introduction

The plasma membranes of eukaryotic cells are organized in an asymmetric manner; at rest, polar and charged lipids are sequestered to the inner leaflet by the activity of ATP-driven pumps. Activation of a specialized class of membrane proteins – phospholipid scramblases – causes rapid collapse of this asymmetry and externalization of negatively charged phosphatidylserine molecules. As a result, extracellular signaling networks, controlling processes such as apoptosis, blood coagulation, membrane fusion and repair, are activated (*Pomorski and Menon, 2006*; *Bevers and Williamson, 2016*; *Nagata et al., 2016*). The TMEM16 family of membrane proteins includes phospholipid scramblases and Cl$^-$ channels (*Falzone et al., 2018*), all of which are Ca$^{2+}$-dependent. Notably, TMEM16 scramblases also mediate Ca$^{2+}$-dependent ion transport (*Malvezzi et al., 2013*; *Scudieri et al., 2015*; *Yu et al., 2015*; *Lee et al., 2016*; *Lee et al., 2018*). Prior structural and functional analyses of the fungal nhTMEM16 scramblase from *Nectria haematococca* identified a membrane-exposed hydrophilic groove that serves as the translocation pathway for ions and lipids

(*Brunner et al., 2014*; *Yu et al., 2015*; *Bethel and Grabe, 2016*; *Jiang et al., 2017*; *Lee et al., 2018*). In the TMEM16A channel, this pathway is sealed from the membrane, preventing lipid access while allowing only ion permeation (*Dang et al., 2017*; *Paulino et al., 2017a*; *Paulino et al., 2017b*).

Phospholipid scramblases are unusual membrane proteins in the sense that their environment serves as their substrate. Activation of scramblases results in the fast and passive transbilayer movement of lipids. Thus, scramblases should affect the surrounding membrane to create a conduit between leaflets though which lipids can diffuse. Current models of lipid translocation, based on the structure of the $Ca^{2+}$-bound conformation of the nhTMEM16 scramblase, postulate a mechanism in which lipid headgroups move through the hydrophilic permeation pathway while the tails remain embedded in the membrane core (*Pomorski and Menon, 2006*; *Brunner et al., 2014*). However, little is known about whether or how lipids and $Ca^{2+}$ binding –the physiologic activation trigger– affect the structure of the TMEM16 scramblases. Further, it is not known how these proteins affect the surrounding membrane to enable lipid scrambling. Here, we use cryo-electron microscopy (cryo-EM) and functional experiments to address these questions. We determine the structures of a functionally characterized TMEM16 scramblase, afTMEM16 from *Aspergillus fumigatus*, in lipid nanodiscs in an inactive ($Ca^{2+}$-free) and an active ($Ca^{2+}$-bound) conformation. These structures allow us to define key conformational rearrangements that underlie $Ca^{2+}$-dependent scramblase activation. Additionally, we show that scrambling is inhibited by the lipid C24:0 ceramide (Cer24:0) and determine the 3.6 Å resolution structure of the $Ca^{2+}$-bound afTMEM16/Cer24:0-nanodisc complex. These structures, together with functional experiments, suggest that the $Ca^{2+}$-dependent conformational rearrangements described here allow the scramblase to locally thin the membrane at the opened lipid pathway to facilitate lipid transport.

## Results

### Structure of the afTMEM16 scramblase in a lipid nanodisc

To isolate conformations of afTMEM16 relevant to its scramblase activity, nanodiscs were formed from a 3:1 mixture of POPE:POPG lipids where afTMEM16 mediates lipid scrambling while its non-selective ion transport activity is silenced (*Malvezzi et al., 2013*; *Lee et al., 2016*). We used single-particle cryo-EM to determine the structures of nanodisc-incorporated afTMEM16 in the presence and absence of $Ca^{2+}$ to resolutions of 4.0 and 3.9 Å, respectively (*Figure 1*). In both conditions, afTMEM16 adopts the TMEM16 fold (*Figure 1*; *Figure 1—figure supplement 1–6*), where each monomer in the dimeric protein comprises a cytosolic domain organized into a 6 α-helix (α1-α6)/3 β-strand (β1-β3) ferredoxin fold and a transmembrane region encompassing 10 α-helices (TM1-TM10) (*Figure 1—figure supplement 5*) (*Brunner et al., 2014*; *Dang et al., 2017*; *Paulino et al., 2017a*; *Bushell et al., 2018*). The two monomers are related by a twofold axis of symmetry at the dimer interface, formed by TM10 and the cytosolic domain (*Figure 1E*). The helices near the dimer interface delimit two large hydrophobic cavities, called dimer cavities, which are lined by TM1, TM2 and TM10 from one monomer and TM3, TM5 from the other (*Figure 1E*). In both maps, the C-terminal portion of TM6 and the linker connecting it to the short cytosolic α4 helix are not well-resolved, likely reflecting their mobility within the membrane (*Figure 1A,B*). When processing the data without imposing symmetry between subunits, the two monomers differ in the well-resolved density of the ~22 C-terminal residues of TM6, likely reflecting the asymmetric orientation of the protein within the nanodisc (*Figure 1—figure supplement 2*). The maps generated by signal subtracting the nanodisc and imposing C2 symmetry are of higher resolution and nearly identical to the non-symmetrized (C1) maps, apart from the resolved portions of TM6 (*Figure 1—figure supplements 2* and *3*), and were therefore used for model building.

In the presence of the functionally saturating concentration of 0.5 mM $Ca^{2+}$ (*Malvezzi et al., 2013*), afTMEM16 adopts a conformation similar to that of $Ca^{2+}$-bound nhTMEM16 and hTMEM16K in detergent (*Brunner et al., 2014*; *Bushell et al., 2018*) (*Figure 1—figure supplement 5*). The TM3-6 from each monomer form a peripheral hydrophilic cavity that opens to the membrane with a minimum diameter of ~5 Å. This pathway was proposed to allow entry and translocation of phospholipid headgroups (*Brunner et al., 2014*; *Yu et al., 2015*; *Lee et al., 2016*; *Jiang et al., 2017*; *Lee et al., 2018*; *Malvezzi et al., 2018*) and is analogous to the ion pathway in the TMEM16A

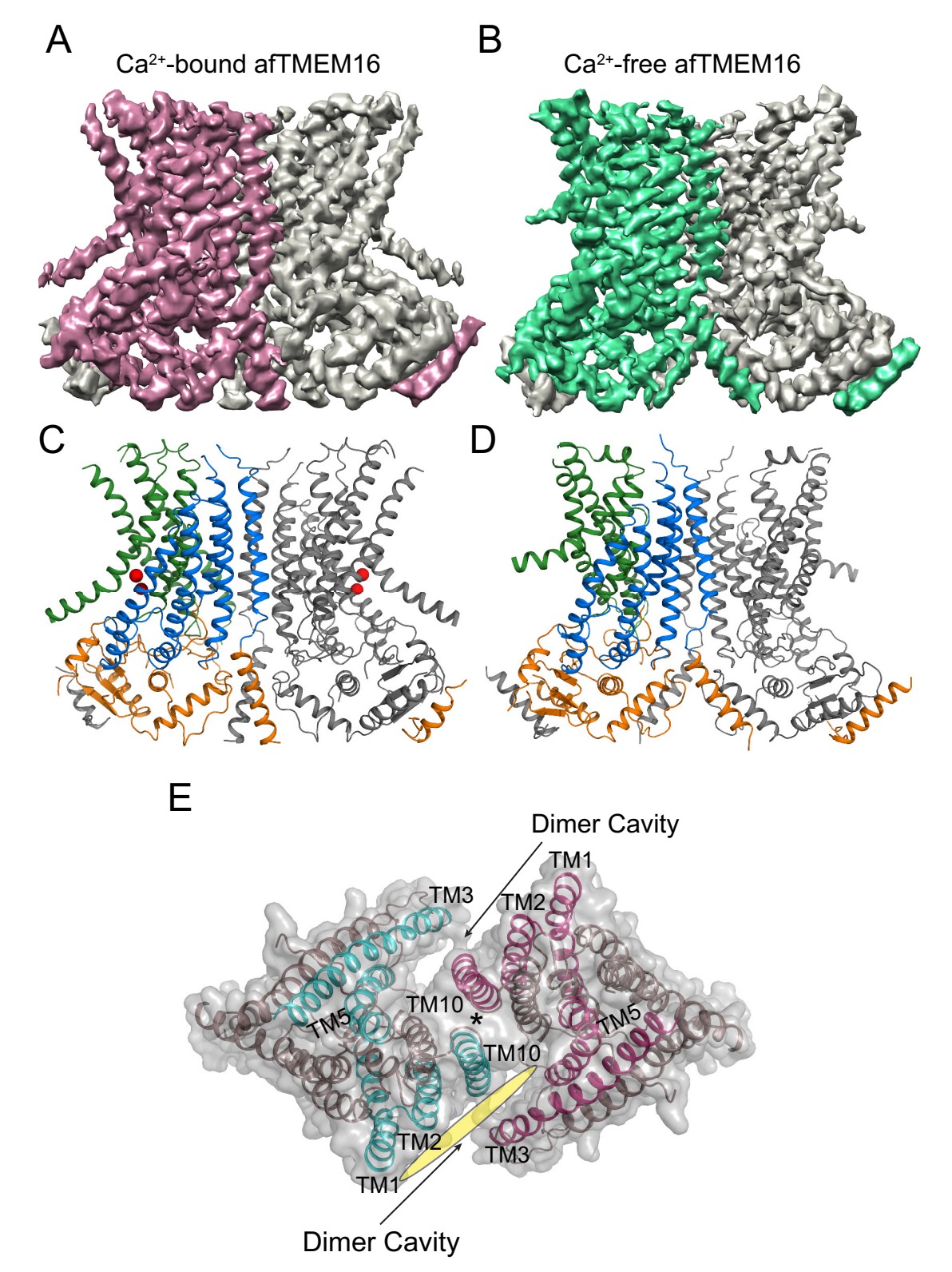

**Figure 1.** Structures of afTMEM16 in the presence and absence of Ca²⁺. (A–B) Masked cryo EM density maps of afTMEM16 in the presence of 0.5 mM Ca²⁺ (A) or in Ca²⁺-free (B) conditions. For clarity, one monomer is shown in gray while the other is colored in red (A, 0.5 mM Ca²⁺) or green (B, 0 Ca²⁺). (C–D) atomic models of afTMEM16 reconstituted in nanodiscs in the presence of 0.5 mM Ca²⁺ (C) and in the absence of Ca²⁺ (D). For clarity one monomer is gray, in the other the cytosolic domain is orange, the permeation pathway is green and the remainder of the protein is blue. Ca²⁺ ions are
*Figure 1 continued on next page*

 Research article

Structural Biology and Molecular Biophysics

*Figure 1 continued*

shown as red spheres. (**D**) Top view of Ca$^{2+}$-bound afTMEM16 shown as maroon ribbon inside its surface representation. The dimer cavities are labeled and one of the two is highlighted with a yellow oval. * denotes the protein's twofold axis of symmetry between the two TM10's. To illustrate the antiparallel orientation of the two dimer cavities, the cavity-lining helices (TM1, 2, 3, 5 and 10) from the two monomers are colored in cyan and red.

DOI: https://doi.org/10.7554/eLife.43229.002

The following figure supplements are available for figure 1:

**Figure supplement 1.** Cryo-EM characterization of afTMEM16/nanodisc complexes.

DOI: https://doi.org/10.7554/eLife.43229.003

**Figure supplement 2.** Asymmetry of afTMEM16-nanodisc complex.

DOI: https://doi.org/10.7554/eLife.43229.004

**Figure supplement 3.** Cryo-EM data processing procedure for afTMEM16/nanodisc complexes.

DOI: https://doi.org/10.7554/eLife.43229.005

**Figure supplement 4.** Representative cryo-EM density for afTMEM16 in 0.5 mM Ca$^{2+}$, in 0 mM Ca$^{2+}$ and in the presence of 0.5 mM Ca$^{2+}$ and 5 mol% C24:0 Ceramide.

DOI: https://doi.org/10.7554/eLife.43229.006

**Figure supplement 5.** Helical organization of afTMEM16.

DOI: https://doi.org/10.7554/eLife.43229.007

**Figure supplement 6.** afTMEM16 in nanodiscs is very similar to nhTMEM16 and human TMEM16K in detergent.

DOI: https://doi.org/10.7554/eLife.43229.008

channel (*Lim et al., 2016*; *Dang et al., 2017*; *Paulino et al., 2017a*; *Paulino et al., 2017b*; *Peters et al., 2018*). Thus, the presence of a lipid environment does not affect the Ca$^{2+}$-bound conformation adopted by the scramblase.

## Ca$^{2+}$ binding induces global rearrangements in the afTMEM16 scramblase

To understand the changes that occur upon Ca$^{2+}$ activation, we compared our Ca$^{2+}$-bound and Ca$^{2+}$-free structures of afTMEM16 and identified global conformational rearrangements of the lipid pathway, of the cytosolic domains and of the Ca$^{2+}$ binding site (*Figure 2*, *Video 1*). In the absence of Ca$^{2+}$, the cytosolic domains of afTMEM16 translate ~3 Å parallel to the plane of the membrane, away from the twofold axis of symmetry, such that the overall cytosolic domain becomes more

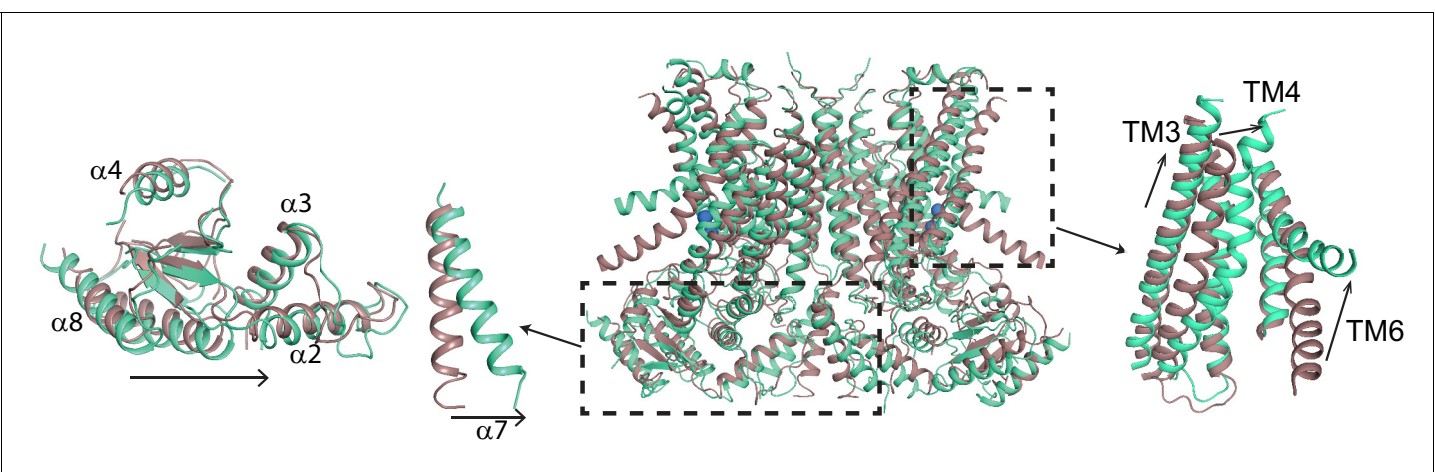

**Figure 2.** Ca$^{2+}$-induced changes in afTMEM16. Structural alignment of afTMEM16 in the presence of 0.5 mM Ca$^{2+}$ (maroon) and absence of Ca$^{2+}$ (cyan) (blue sphere). Left: conformational changes in the cytosolic domain, Right: conformational changes in the lipid permeation pathway. Arrows indicate direction of movement from the Ca$^{2+}$-bound to the Ca$^{2+}$-free conformations.

DOI: https://doi.org/10.7554/eLife.43229.009

The following figure supplement is available for figure 2:

**Figure supplement 1.** afTMEM16 dimer interface.

DOI: https://doi.org/10.7554/eLife.43229.010

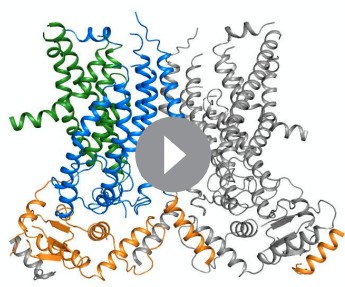

**Video 1.** Ca²⁺-activation of afTMEM16. Morph between Ca²⁺-free and Ca²⁺-bound conformations of afTMEM16. For clarity one monomer is shown in gray. In the other, the cytosolic domain is in orange, the lipid permeation pathway (TM3-7) in green, and the rest of the protein in blue (TM1-2 and TM8-10). Note the downward motion of TM6 and movement of the cytosolic domains parallel to the membrane plane.
DOI: https://doi.org/10.7554/eLife.43229.014

expanded and the cytosolic α7 helices tilt toward the axis of symmetry with no noticeable movement of the transmembrane dimer interface (*Figure 2*, left panel; *Figure 2—figure supplement 1*). In the Ca²⁺-free conformation, the afTMEM16 lipid pathway is closed to the membrane by a pinching motion of the extracellular portions of TM4 and TM6, which move toward each other by ~7 and~3 Å, respectively (*Figure 2*, right panel). From the Ca²⁺-bound conformation, TM4 bends around two prolines (P324 and P333) and TM3 slides by ~6 Å to reach the Ca²⁺-free conformation (*Figure 3A*). Additionally, the intracellular portion of TM6 kinks around A437 by ~20°, inducing a ~ 16 Å vertical displacement of its terminal end (*Figure 3A*, *Video 2*). These rearrangements lead to tighter packing of side chains from TM4 and TM6 and result in exposure of a hydrophobic surface to the membrane core (*Figure 3B*). The pathway is also closed to ion entry by multiple stacked aromatic and hydrophobic side chains from TM3-7 (*Figure 3C*). The narrowest access point of the lipid pathway constricts from ~5 to ~1 Å in the absence of Ca²⁺ preventing lipid entry (*Figure 3D–F*). Notably, mutating residues at the interface between helices that rearrange upon Ca²⁺ binding results in severely impaired scrambling activity in the closely related nhTMEM16 homologue (*Jiang et al., 2017*; *Lee et al., 2018*) (*Figure 3—figure supplement 1*), highlighting the importance of these dynamic rearrangements. The global Ca²⁺-dependent rearrangements of the afTMEM16 scramblase contrast to the local conformational changes seen in the TMEM16A channel, where only TM6 bends upon Ca²⁺ions leaving the binding sites (*Figure 3—figure supplement 2*) (*Paulino et al., 2017a*).

## Ca²⁺-dependent conformational changes of the regulatory Ca²⁺-binding sites

The transmembrane region of afTMEM16 contains two Ca²⁺-binding sites, located between TM6, TM7, and TM8 (*Figure 4A,B*). In the presence of 0.5 mM Ca²⁺, both sites are occupied by calcium ions, as evidenced by the strong densities in the cryo-EM map and in an 'omit' difference map, calculated between experimental data and simulated maps not containing Ca²⁺ (*Figure 4A*). The bound Ca²⁺ ions are coordinated by five conserved acidic residues (E445 on TM6, D511 and E514 on TM7, and E534 and D547 on TM8), three polar residues (Q438, Q518, N539), and the unpaired main chain carbonyl of G441 (*Figure 4A–C*). This coordination is similar to that seen in the nhTMEM16 and hTMEM16K scramblases as well as in the TMEM16A channel, consistent with the evolutionary conservation of the Ca²⁺-binding residues (*Figure 4C*) (*Yu et al., 2012*; *Malvezzi et al., 2013*; *Terashima et al., 2013*; *Brunner et al., 2014*; *Tien et al., 2014*; *Lim et al., 2016*; *Dang et al., 2017*; *Paulino et al., 2017a*; *Bushell et al., 2018*). In the absence of Ca²⁺, the binding site is disrupted by the movement of TM6 which displaces the three residues participating in the site (Q438, G441 and E445) (*Figure 4D–F*, *Video 3*). Additional rearrangements of TM8, displacing N539 and E543, further disrupt the Ca²⁺-binding site (*Figure 4D–F*, *Video 3*). No Ca²⁺ density was visible in the cryo-EM map (*Figure 4E*), confirming that the scramblase is in a Ca²⁺-free conformation. The movement of TM6 and TM8 in opposite directions partially relieves the electrostatic repulsion of the uncoordinated acidic side chains that form the Ca²⁺-binding site and opens a wide, water-accessible, conduit for ions to access the binding site from the intracellular solution.

## Structural analysis of the afTMEM16/nanodisc complex

Because these structures are of protein/nanodisc complexes, they also provide insights into the Ca²⁺-dependent interactions between the scramblase and the nanodisc membrane (*Figure 5*).

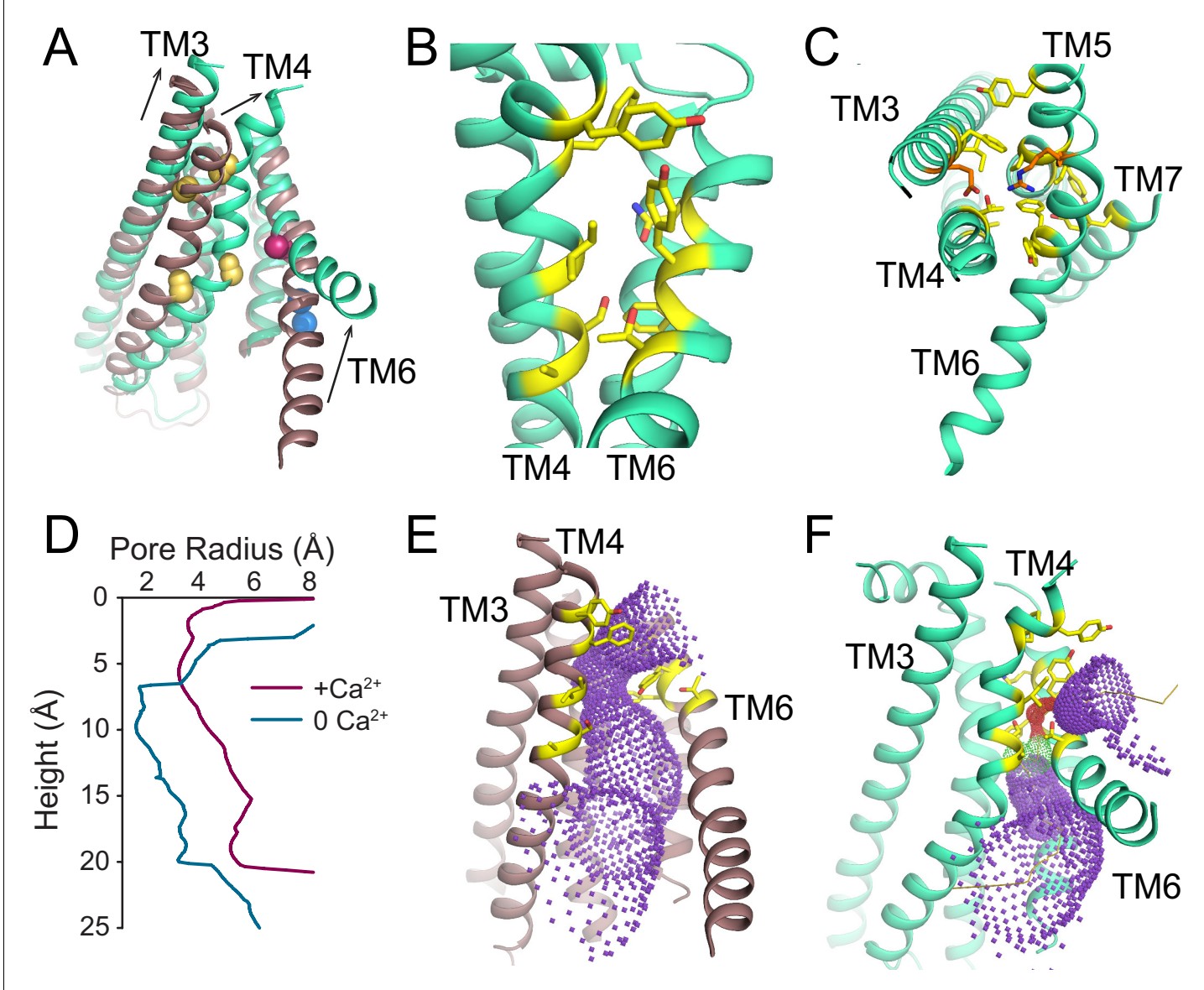

**Figure 3.** Ca$^{2+}$-dependent rearrangements of the lipid permeation pathway. (**A**) Structural alignment of the lipid pathway with afTMEM16 in the presence (maroon) or absence (cyan) of 0.5 mM Ca$^{2+}$. The color scheme is the same throughout the figure. Arrows indicate direction of movement from the Ca$^{2+}$-free to the Ca$^{2+}$-bound conformations. The lipid permeation pathway is constricted by rearrangements of TM4 around P324 and P333 (shown as yellow spheres in both structures) and TM6 at A437 (shown as a red sphere in both structures). (**B**) Close-up view of the closed permeation pathway, residues at the interface with the membrane are shown as yellow sticks. (**C**) Top view of the permeation pathway in the absence of Ca$^{2+}$. Residues pointing inside the pathway are shown as yellow sticks. The interacting charged pair E305 and R425 are shown as orange sticks. (**D**) Diameter of the afTMEM16 lipid pathway in the presence (maroon) and absence (cyan) of Ca$^{2+}$. The diameter was estimated using the HOLE program (*Smart et al., 1996*). (**E–F**) Accessibility of the lipid permeation pathway estimated using the program HOLE in the presence (**E**) or absence (**F**) of Ca$^{2+}$. Purple denotes areas of diameter d > 5.5 Å, yellow areas where 5.5 < d < 2.75 Å and red areas with d < 2.75 Å.

DOI: https://doi.org/10.7554/eLife.43229.011

The following figure supplements are available for figure 3:

**Figure supplement 1.** Residues important for scrambling are mapped onto afTMEM16 open and closed permeation pathways.

DOI: https://doi.org/10.7554/eLife.43229.012

**Figure supplement 2.** Ca$^{2+}$-induced changes in TMEM16A.

DOI: https://doi.org/10.7554/eLife.43229.013

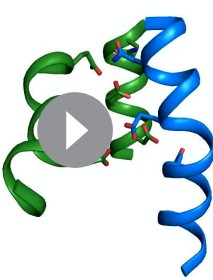

**Video 2.** Ca$^{2+}$-dependent movement of the afTMEM16 lipid pathway. Close-up view of the rearrangements undergone by the afTMEM16 lipid pathway (TM3-7) upon Ca$^{2+}$ binding. Note that upon Ca$^{2+}$ binding TM4 straightens and moves out of the pathway, TM3 slides downward and TM6 moves out of the pathway and bends downwards to open the pathway.
DOI: https://doi.org/10.7554/eLife.43229.015

Inspection of these maps reveals that the nanodiscs containing afTMEM16 are bent along the two dimer cavities of the scramblase (*Figure 5A, C*) and that there is a region of low density at the open lipid pathway (*Figure 5B*). The nanodisc density is a convolution of the lipid molecules and the MSP1E3 protein that defines the nanodisc boundary, suggesting that presence of the afTMEM16 scramblase affects the orientation of both. The bending along the dimer cavities of afTMEM16 is seen in the 3D reconstructions from all datasets presented here, suggesting that it does not depend on the presence of Ca$^{2+}$, on the conformation of the protein or processing algorithm (*Figure 5A,C*, *Figure 5—figure supplement 1–3*, *Supplementary file 1*). Remarkably, bending of the nanodisc is also visible in 2D classes of particles containing afTMEM16, but is absent from protein-free ones (*Figure 5—figure supplement 4*). In all maps, the membrane appears to be thicker around the long TM3 helix in one monomer and thinner at the short TM1 of the other (*Figure 5E*). This bending is visible along both dimer cavities, irrespective of the positioning of the scramblase within the nanodisc (*Figure 5A,C*, *Figure 5—figure supplement 1*). The independence of the observed membrane bending on Ca$^{2+}$ is consistent with the lack of Ca$^{2+}$-dependent rearrangements undergone by these cavities (*Figure 3A*). The membrane slant matches the tilted plane defined by the extracellular ends of the five, dimer cavity-lining helices (*Figure 5E*) suggesting that it is caused by the architecture of afTMEM16.

In contrast to the membrane bending, the region of weaker density of the nanodisc membrane near the lipid pathway depends on the scramblase conformation (*Figure 5B,D*). In the three datasets collected in the presence of 0.5 mM Ca$^{2+}$, the density between the TM4 and TM6 helices that delimit the open lipid transport pathway is weaker than in the rest of the complex (*Figure 5B*, insets, *Figure 5—figure supplement 2*). This weakening can be seen in both subunits irrespective of the distance of the pathway from the nanodisc edge (*Figure 5B*, insets). In the absence of Ca$^{2+}$, no region of weak density is visible (*Figure 5D*, inset) as the conformational rearrangement of TM4 and TM6 pinches shut the permeation pathway. The weaker density suggests that the membrane is thinner and/or more disordered near the open lipid pathway of an active scramblase.

## Lipid dependence of scrambling by afTMEM16

The hypothesis that afTMEM16 alters the membrane to scramble lipids suggests that scrambling should be impaired by thicker membranes and by bilayer modifying lipids. To increase the membrane thickness, we reconstituted afTMEM16 into liposomes composed of lipids of varying acyl chain length and saturation. The rate of scrambling was determined with a dithionite-based fluorescence assay (*Figure 6A*) (*Malvezzi et al., 2013*; *Malvezzi et al., 2018*). We found that afTMEM16 is equally active in liposomes formed from mixtures of POPE:POPG or POPC:POPG lipids, which have C16:0 and C18:1 acyl chains, and in liposomes formed from DOPC:DOPG lipids, where both acyl chains are C18:1 (*Figure 6B*, *Figure 6—figure supplement 1*, *Supplementary file 2*). This suggests that the saturation of the acyl chains does not influence afTMEM16 scrambling activity. In contrast, increasing membrane thickness by ~7 Å by forming liposomes with DEPC:DEPG lipids (*Lewis and Engelman, 1983*), with C22:1 acyl chains, reduces the scrambling rate of afTMEM16 by ~500 fold in the presence of Ca$^{2+}$ and ~20-fold in the absence of Ca$^{2+}$ (*Figure 6A–B*, *Figure 6—figure supplement 1*, *Supplementary file 2*). The observation that scrambling is inhibited in thicker membranes is consistent with our hypothesis that afTMEM16 thins the membrane to scramble lipids.

Among naturally occurring bilayer-modifying constituents of cellular membranes, we focused our attention on ceramides, as these sphingolipids regulate cellular processes that involve activation of phospholipid scramblases, such as blood coagulation, inflammation and apoptosis (*Deguchi et al.,*

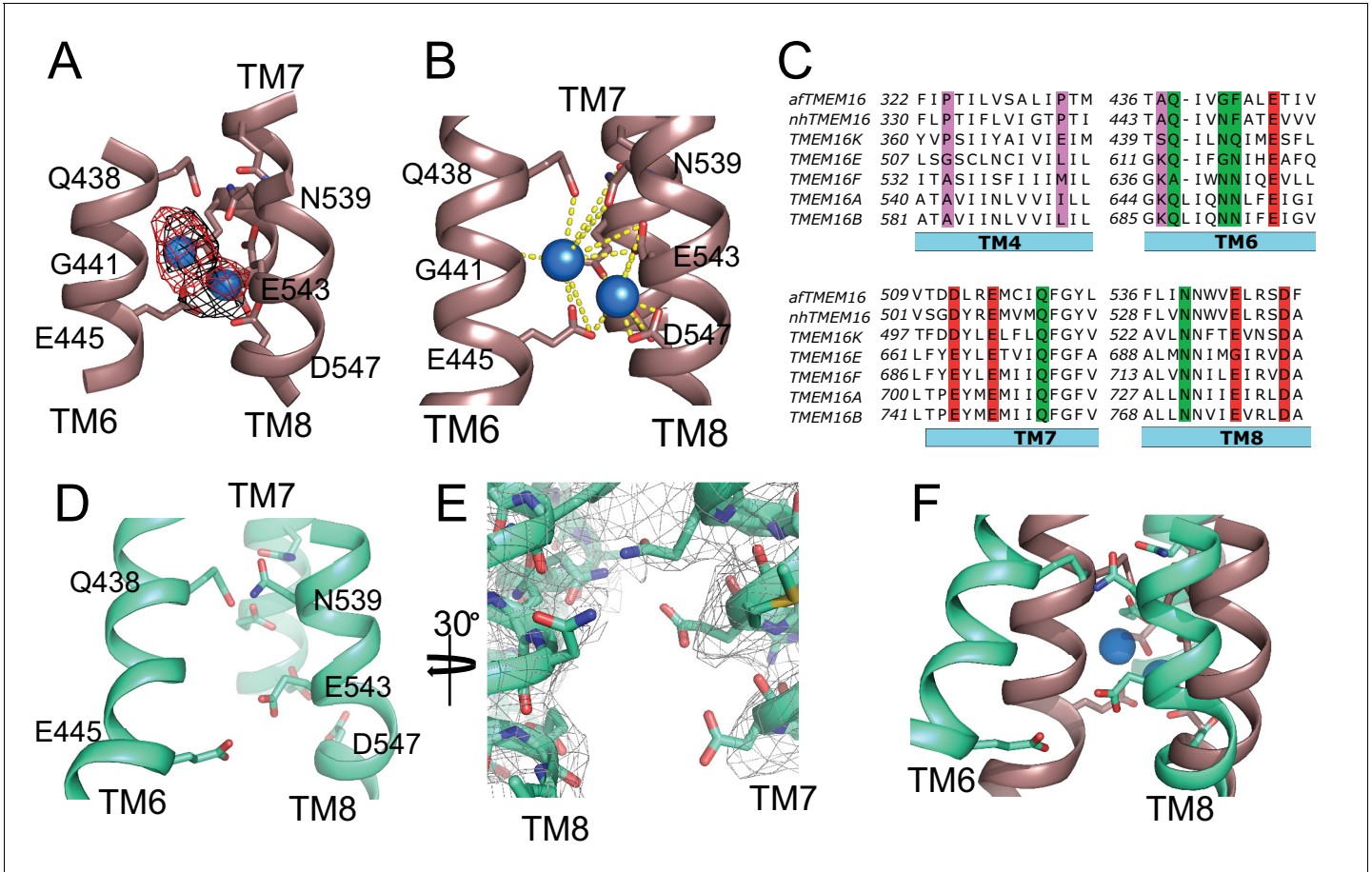

**Figure 4.** afTMEM16 $Ca^{2+}$-binding site and conformational changes. (A) Close up view of the $Ca^{2+}$-binding site, with key coordinating residues shown as sticks. The density corresponding to the $Ca^{2+}$ ions (blue spheres) from the experimental map is shown in black and the density from the calculated omit difference map is shown in red. The peak density corresponding to the $Ca^{2+}$ ions in the omit difference maps (red mesh) is $\sigma = 13$ and 7. (B) $Ca^{2+}$ coordination in afTMEM16. (C) Structure-based sequence alignment of the $Ca^{2+}$-binding site and gating region of TMEM16 proteins. The alignment was generated using PROMALS3D (*Pei and Grishin, 2014*). Highlighted residues: conserved acidic (red) or polar (green) side chain in $Ca^{2+}$ binding site, and the residues around which TM4 and TM6 bend (pink). (D) Close-up of the $Ca^{2+}$-binding site in the absence of ligand. Coordinating residues on TM6 and TM8 are labeled. (E) EM density of residues lining the $Ca^{2+}$-binding site in the absence of $Ca^{2+}$ highlighting the lack of density for $Ca^{2+}$ ions. (F) Structural alignment of the binding site with and without $Ca^{2+}$.

DOI: https://doi.org/10.7554/eLife.43229.016

2004; *Hannun and Obeid, 2008*; *Borodzicz et al., 2015*; *Cantalupo and Di Lorenzo, 2016*; *Deguchi et al., 2017*). We found that addition of physiological levels of long chain ceramides potently inhibits scrambling by reconstituted afTMEM16 (*Figure 6C*, *Figure 6—figure supplement 2A–H*). Among the tested ceramides, C24:0 Ceramide (Cer24:0) inhibits scrambling ~250 fold when added at 5 mole% (*Figure 6C*). The inhibitory effect depends on ceramide concentration, with minimal effects at 1 mole%, as well as acyl chain saturation, as Cer24:1 is nearly inert (*Figure 6C*, *Figure 6—figure supplement 2*). The length of the ceramide acyl chain is also important as Cer22:0 is as potent as Cer24:0 while Cer18:0 has minimal effect at 5 mole% (*Figure 6C*, *Figure 6—figure supplement 2F*). The long chain ceramides also inhibit scrambling in the absence of $Ca^{2+}$ by ~10-fold (*Figure 6—figure supplement 2G*). The inhibitory effects of ceramides or thicker bilayers do not reflect impaired reconstitution of the protein (*Figure 6—figure supplement 2H*). We used a gramicidin-based fluorescence quench assay (*Ingólfsson and Andersen, 2010*) to investigate whether Cer24:0 and Cer24:1 differentially affect bulk membrane properties, such as thickness and fluidity. The assay monitors alterations in the gramicidin monomer↔dimer equilibrium, which varies with changes in membrane thickness and elasticity (*Andersen et al., 2007*; *Lundbæk et al., 2010*).

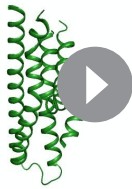

**Video 3.** Ligand-dependent rearrangements of the Ca²⁺ binding site. Close-up view of the rearrangements undergone by the afTMEM16 ligand binding site upon Ca²⁺ binding. TM6 and TM7 are shown in green and TM8 is shown in blue. The residues directly involved in Ca²⁺ coordination are shown as sticks. Upon Ca²⁺ binding, TM6 moves toward TM7 and TM8 while TM8 tilts back toward TM7 to form the binding site.
DOI: https://doi.org/10.7554/eLife.43229.017

Addition of Cer24:0 or Cer24:1 comparably reduces gramicidin activity (*Figure 6—figure supplement 2I–K*), indicating that both ceramides stiffen and/or thicken the membrane to a similar extent. The comparable effects of Cer24:0 and Cer24:1 on membrane properties suggest that their distinct effects on afTMEM16 scrambling might reflect specific interactions with the scramblase and/or their differential ability to form gel-like microdomains in membranes (*Pinto et al., 2008*; *García-Arribas et al., 2017*; *Alonso and Goñi, 2018*).

## Structure of the Ca²⁺-bound and ceramide inhibited afTMEM16/ nanodisc complex

To understand how long-tail ceramides affect scrambling, we determined the 3.6 Å resolution structure of Ca²⁺-bound afTMEM16 in nanodiscs containing 5 mole% Cer24:0 (*Figure 7*; *Figure 1— figure supplement 1–4*, *Figure 7—figure supplement 1*), a concentration that drastically inhibits activity (*Figure 6C*). The protein adopts a conformation where the lipid pathway is open to the membrane, both Ca²⁺-binding sites are occupied, and is nearly identical to the Ca²⁺-bound active state, with an overall Cα r.m.s.d. <1 Å (*Figure 7A*). No individual lipids are resolved within the pathway, while several densities that we attributed to partial acyl chains are visible in the dimer cavity (*Figure 7—figure supplement 1*). In the Cer24:0-inhibited structure, the nanodisc membrane bends along the dimer cavities in a similar manner to what was seen in the Ca²⁺-bound and Ca²⁺-free structures, (*Figure 7B*). Interestingly, the density near the lipid pathway in the presence of Cer24:0 appears to be less weakened (*Figure 7C*) than that seen in the Ca²⁺-bound structure (*Figure 5B*), even though the pathway is open in both structures. While direct comparisons of the densities between the two structures are difficult because of their different resolutions, it is tempting to speculate that the density of the nanodisc membrane in this area correlates with the activity of the protein. Further work will be required to test this possibility. The finding that afTMEM16 adopts a Ca²⁺-bound conformation with open lipid pathways suggest that long chain ceramides do not inhibit scrambling by preventing the Ca²⁺-dependent opening conformational transition. Rather, our findings suggest that the physico-chemical properties of the membrane determine whether lipid scrambling actually occurs.

## Discussion

Despite recent advances (*Falzone et al., 2018*), the molecular mechanisms underlying the Ca²⁺-dependent activation of the TMEM16 scramblases and their interactions with the surrounding membrane lipids remain poorly understood. Here, we use cryo electron microscopy to determine the structure of a functionally well-characterized TMEM16 family member, afTMEM16 (*Malvezzi et al., 2013*; *Lee et al., 2016*; *Malvezzi et al., 2018*), in a membrane environment. Our results show that the afTMEM16 scramblase reconstituted in lipid nanodiscs undergoes global conformational rearrangements upon Ca²⁺ binding (*Figures 1–4*). These rearrangements involve the closure of the pathway via a pinching motion of TM4 and TM6, the upward motion of TM3 and the dilation of the Ca²⁺-binding site. The global nature of these rearrangements differs greatly from those seen in the TMEM16A channel, where only TM6 moves in response to Ca²⁺ binding (*Dang et al., 2017*; *Paulino et al., 2017a*) (*Figure 3—figure supplement 2*). The movements we observe in afTMEM16, bear striking similarities to those predicted by MD simulations for the nhTMEM16 scramblase upon removal of Ca²⁺ (*Jiang et al., 2017*). Notably, similar rearrangements are observed in the human TMEM16K scramblase (*Bushell et al., 2018*), supporting their evolutionary conservation.

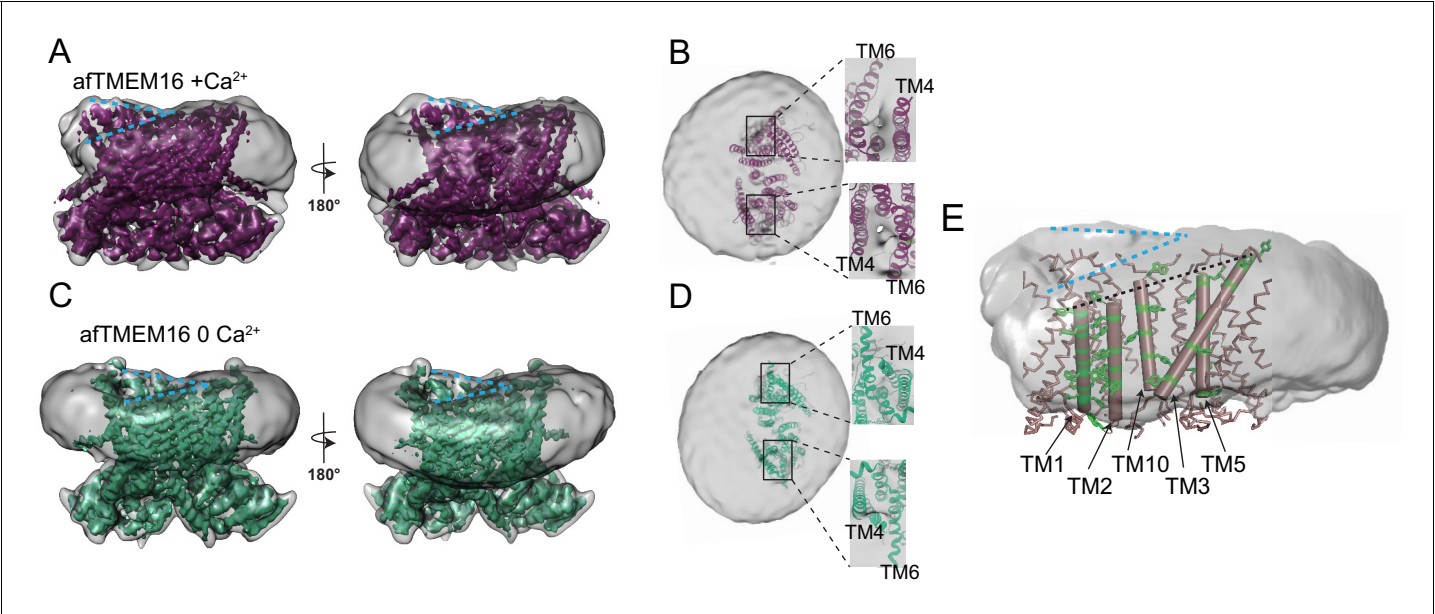

**Figure 5.** afTMEM16 affects the surrounding membrane. (**A and C**) The unmasked maps (gray) of the afTMEM16/nanodisc complex in the presence (**A**) and absence of Ca$^{2+}$ (**C**) are shown low pass filtered to 10 Å at σ = 0.4 with front (left panels) and back (right panels) views. The masked map of the afTMEM16 scramblase in 0.5 mM Ca$^{2+}$ (**A**), in purple) or in 0 Ca$^{2+}$ (**C**), in green) are shown inside the respective unmasked maps. Blue dashed lines highlight the bending at the dimer cavities. (**B and D**) Top view of the unmasked map (grey) with (**B**) and without (**D**) Ca$^{2+}$ low-pass filtered to 10 Å at σ = 0.1 with the respective atomic models shown as ribbons inside. Insets show close-up views of the nanodiscs density at the permeation pathways. Note that the insets are displayed at a tilted angle compared to the top views to visualize the lipid pathway. (**E**) Bending of the outer leaflet at the dimer cavities in the presence of 0.5 mM Ca$^{2+}$. The low-pass filtered map of the complex (from **A**) was segmented and the isolated nanodisc density is shown. The afTMEM16 transmembrane region is shown in ribbon with the helices lining one dimer cavity (TM1, 2, 3, 5, and 10) in cartoon representation. Aromatic residues are shown as green sticks. Blue dashed lines trace the upper membrane leaflet at the two sides of the lipid permeation pathway highlighting the opposite slant. Black dashed line traces the slope of the helices lining the dimer cavity.

DOI: https://doi.org/10.7554/eLife.43229.018

The following figure supplements are available for figure 5:

**Figure supplement 1.** Ca$^{2+}$-dependent and Ca$^{2+}$-independent effects of afTMEM16 on nanodisc density.
DOI: https://doi.org/10.7554/eLife.43229.019

**Figure supplement 2.** The effects of Ca$^{2+}$-bound afTMEM16 on the surrounding membrane are seen in three independent datasets with varying resolutions.
DOI: https://doi.org/10.7554/eLife.43229.020

**Figure supplement 3.** The effects of Ca$^{2+}$-bound afTMEM16 on the surrounding membrane do not depend on the processing algorithm.
DOI: https://doi.org/10.7554/eLife.43229.021

**Figure supplement 4.** Altered membrane organization induced by activation of afTMEM16.
DOI: https://doi.org/10.7554/eLife.43229.022

Our structural and functional experiments provide detailed insights into the activation mechanism of TMEM16 phospholipid scramblases (*Figures 1–4*) and how these proteins alter the surrounding membrane to facilitate the transfer of lipids between leaflets (*Figures 5–7*). The Ca$^{2+}$-free and Ca$^{2+}$-bound structures of afTMEM16 define the extremes of the Ca$^{2+}$-dependent activation process and suggest that opening of the lipid pathway is primarily controlled by two structural elements, TM4 and TM6 (*Figure 8*). Without Ca$^{2+}$, TM4 and TM6 are bent, sealing the pathway from the lipid membrane (*Figure 8A*). The first step in activation is presumably Ca$^{2+}$ binding, which facilitates the transition of TM6 to a straight conformation and its disengagement from TM4, allowing TM6 to move toward TM8 and complete the formation of the Ca$^{2+}$-binding site (*Figure 8B*). The resulting proposed conformation is similar to that of Ca$^{2+}$-bound TMEM16A, where TM6 is straight but lipid access is prevented by a bent TM4 (*Dang et al., 2017*; *Paulino et al., 2017a*). Notably, an intermediate, Ca$^{2+}$-bound conformation with a closed lipid pathway has been recently observed for the hTMEM16K scramblase (*Bushell et al., 2018*). Straightening of the TM4 helix opens the lipid pathway to enable lipid translocation, as seen in Ca$^{2+}$-bound afTMEM16 and nhTMEM16 (*Figure 8C*)

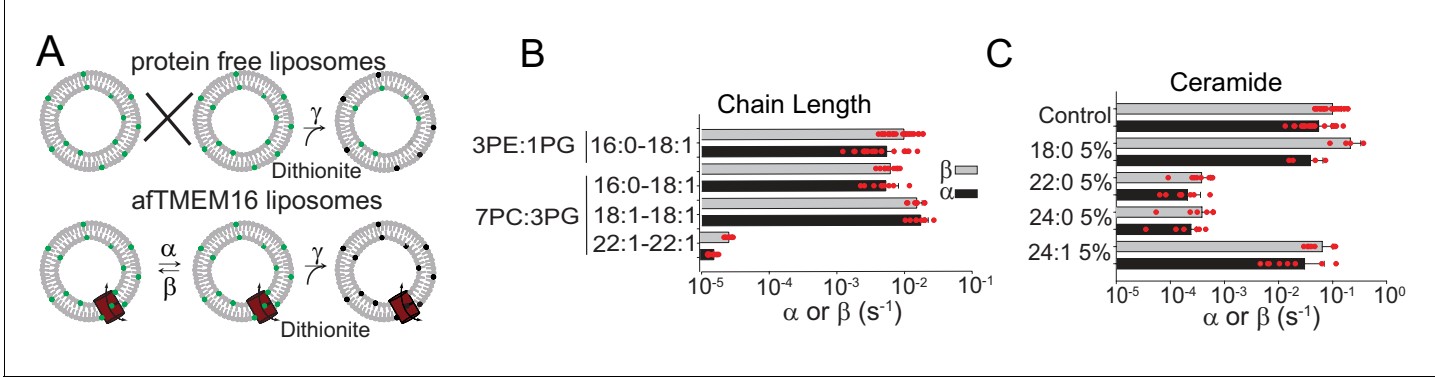

**Figure 6.** Dependence of afTMEM16 on membrane lipids. (**A**) Schematic of the in vitro scramblase assay. Liposomes are reconstituted with NBD-labeled phospholipids (green) that distribute equally in the two leaflets. Addition of extraliposomal sodium dithionite reduces the NBD fluorophore (black), causing 50% fluorescence loss in protein-free vesicles (top panel). When a scramblase is present (bottom panel), all NBD-phospholipids become exposed to dithionite, resulting in complete loss of fluorescence (*Malvezzi et al., 2013*). (**B–C**), Forward ($\alpha$, black) and reverse ($\beta$, grey) scrambling rate constants of afTMEM16 in 0.5 mM $Ca^{2+}$ as a function of lipid chain length (**B**) or addition of 5 mole% of different ceramides (**C**). Rate constants were determined by fitting the fluorescence decay time course to *Eq. 1*. For the chain length experiments liposomes were formed either from a 3 POPE: 1 POPG or a 7 POPC: 3 POPG mixture (**B**). A 3 POPE: 1 POPG mixture was used for the reconstitutions containing ceramide lipids (**C**). Data is reported as mean ±S.D. Red circles denote individual experiments. Values and exact repeat numbers are reported in *Supplementary files 2,4*.

DOI: https://doi.org/10.7554/eLife.43229.023

The following figure supplements are available for figure 6:

**Figure supplement 1.** Functional modulation of lipid scrambling by acyl chain length.

DOI: https://doi.org/10.7554/eLife.43229.024

**Figure supplement 2.** Functional modulation of lipid scrambling by addition of ceramide lipids.

DOI: https://doi.org/10.7554/eLife.43229.025

(*Brunner et al., 2014*). To complete the gating scheme, we propose a state where TM4 is straight and TM6 is bent (*Figure 8D*), which would give rise to a partially opened lipid pathway. This conformation, while not yet observed experimentally, would account for the low, basal activity of

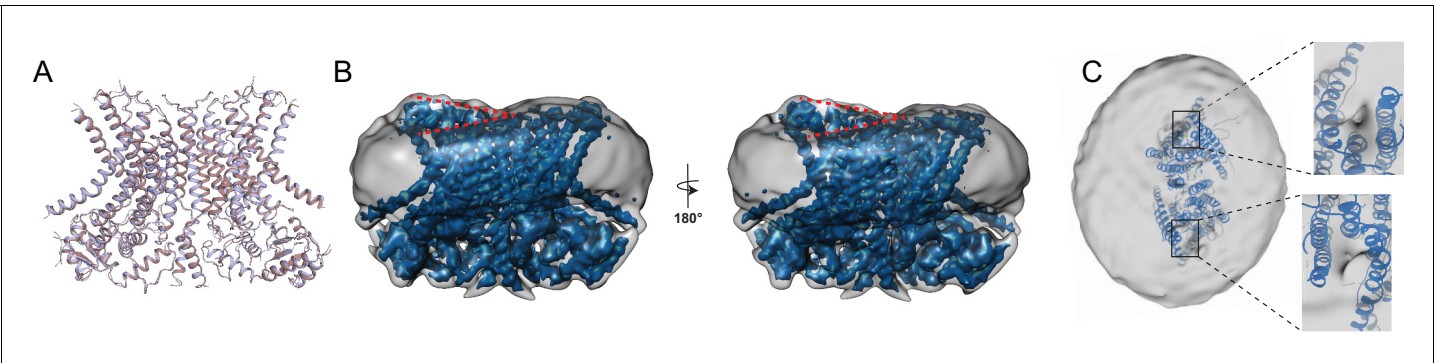

**Figure 7.** Structure of afTMEM16 in the presence of C24:0 ceramide. (**A**) Structural alignment of $Ca^{2+}$-bound afTMEM16 with (light blue) and without C24:0 ceramide (maroon) with a C$\alpha$ r.m.s.d. <1 Å. (**B**) The unmasked maps (gray) of the afTMEM16/nanodisc complex in the presence of 0.5 mM $Ca^{2+}$ and 5 mol% C24:0 ceramide is shown low pass filtered to 10 Å at $\sigma$ = 0.4 with front (left panel) and back (right panel) views. The masked map of the afTMEM16 scramblase in 0.5 mM $Ca^{2+}$ and 5 mol% C24:0 ceramide is shown inside the respective unmasked maps. Red dashed lines highlight the membrane bending. (**C**) Top view of unmasked map (grey) low pass filtered to 10 Å at $\sigma$ = 0.1 with the atomic models inside. Insets show the density at the permeation pathway. Note that the insets are displayed at a tilted angle compared to the top views to visualize the changes in density at the lipid pathway.

DOI: https://doi.org/10.7554/eLife.43229.026

The following figure supplement is available for figure 7:

**Figure supplement 1.** Lipid-like density in the dimer cavity.

DOI: https://doi.org/10.7554/eLife.43229.027

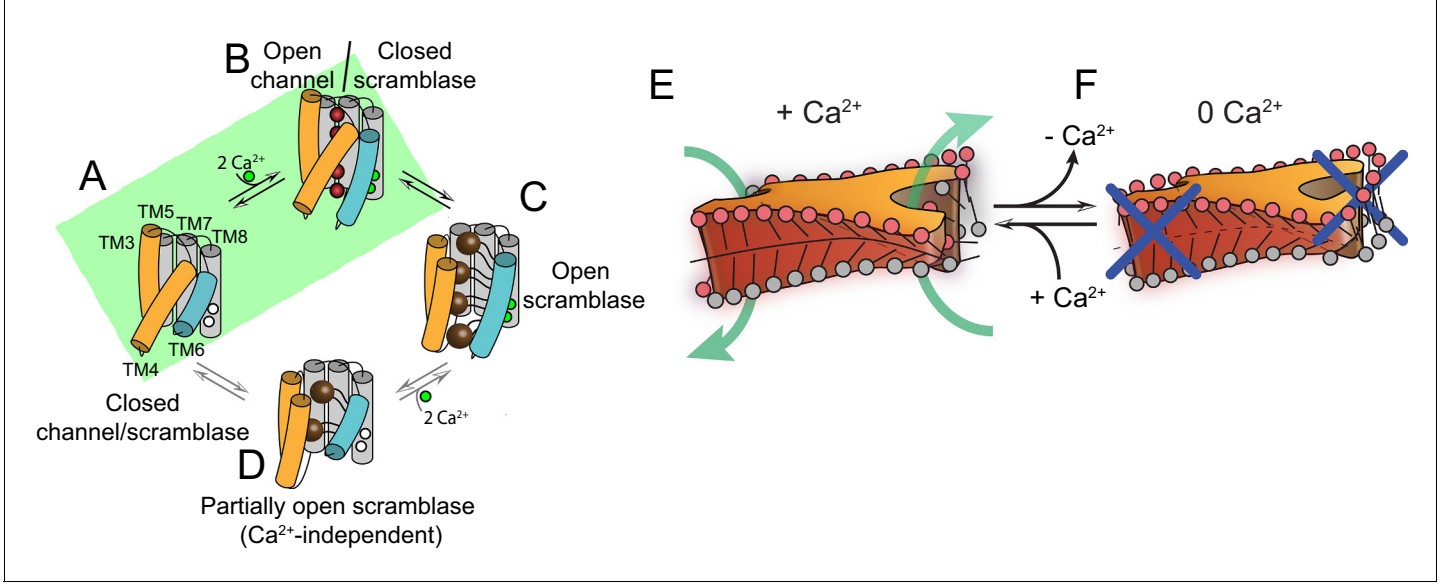

**Figure 8.** Proposed mechanisms for gating and membrane remodeling by TMEM16 scramblases. (**A–D**) $Ca^{2+}$-dependent gating scheme for TMEM16 scramblases. The α helices lining the lipid pathway and $Ca^{2+}$-binding site (TM3-8) are shown as cylinders. The two gating elements (TM3-4 and TM6) are colored in orange and blue, respectively. When the $Ca^{2+}$ binding sites are empty TM4 and TM6 are bent and occlude the pathway, resulting in a closed scramblase (PDBID: 6DZ7) (**A**). Upon $Ca^{2+}$ binding, TM6 straightens and partly disengages from TM4, giving rise to a pathway that is closed to lipids but that can potentially allow ion permeation (a mTMEM16A-like state, PDBID:5OYG) (**B**). Rearrangement of TM6 promotes the straightening of TM4, resulting in an open lipid pathway (PDBID: 4WIS, 6E0H) (**C**). Straightening of TM4 in the absence of $Ca^{2+}$, gives rise to a partially open lipid pathway that might allow the experimentally observed $Ca^{2+}$-independent lipid scrambling (*Malvezzi et al., 2013*) (**D**). If the straightening of TM4 is energetically unfavorable, the protein is restricted to visiting only states (**A**) and (**B**), green shaded area. This would give rise to the observed $Ca^{2+}$-dependent gating behavior of the TMEM16 channels. (**E–F**) proposed mechanism for membrane remodeling by the afTMEM16 dimer. The slanted architecture of the dimer cavity primes the membrane by bending it in opposite directions on the two sides of the lipid translocation pathway. In the presence of $Ca^{2+}$ the pathway is open, enabling the formation of a membrane area that is thin and forming a conduit through which lipid headgroups can translocate between the two leaflets (**E**). In the absence of $Ca^{2+}$ the pathway is closed preventing lipid scrambling (**F**).
DOI: https://doi.org/10.7554/eLife.43229.028

The following figure supplements are available for figure 8:

**Figure supplement 1.** Structural comparison of $Ca^{2+}$-bound and $Ca^{2+}$-free afTMEM16 and TMEM16A.
DOI: https://doi.org/10.7554/eLife.43229.029

**Figure supplement 2.** Architecture of the dimer cavity of TMEM16 proteins.
DOI: https://doi.org/10.7554/eLife.43229.030

**Figure supplement 3.** Effects of afTMEM16 on the nanodsic membrane are not observed in other unrelated membrane proteins.
DOI: https://doi.org/10.7554/eLife.43229.031

reconstituted scramblases in the absence of $Ca^{2+}$ (*Figure 6—figure supplements 1* and *2*) (*Malvezzi et al., 2013*; *Brunner et al., 2014*; *Lee et al., 2016*; *Lee et al., 2018*; *Malvezzi et al., 2018*). Within this gating mechanism, $Ca^{2+}$-activated TMEM16 channels would naturally arise from scramblases via mutations that render straightening of TM4 unfavorable, while maintaining the $Ca^{2+}$-dependent rearrangement of TM6 (*Figure 8A,B*) (*Dang et al., 2017*; *Paulino et al., 2017a*). Indeed, in the $Ca^{2+}$-free afTMEM16 scramblase, the pathway and cytosolic domain adopt conformations similar to those seen in the TMEM16A channel (*Figure 8—figure supplement 1*). Mutations that convert TMEM16A into a scramblase (*Yu et al., 2015*; *Jiang et al., 2017*) might re-enable the TM4 transition.

Our structures of afTMEM16 scramblase/nanodisc complexes suggest that lipid scrambling is enabled by two features of the TMEM16 architecture: the dimer cavities and the lipid pathway (*Figures 5* and *7*). The helices lining the dimer cavity are of different heights; TM3 and TM5 of one monomer are longer than the TM1 and TM2 from the other (*Figure 5E*) and the upper leaflet of the membrane bends to track this sloping. The shortest helices (TM1 and TM2) are unusually rich in aromatic side chains, which anchor TM segments to membrane/solution interfaces (*O'Connell et al., 1990*), creating a favorable environment for both phospholipid headgroups and tails (*Figure 5E*,

*Figure 8—figure supplement 2*). This architecture is conserved in the nhTMEM16 and hTMEM16K scramblases (*Figure 8—figure supplement 2A,B*). In contrast, in the mTMEM16A channel TM1 is longer and contains fewer aromatics, and TM3 is shorter, such that the extracellular termini of these helices sit at comparable heights within the membrane (*Figure 8—figure supplement 2C,D*). The lesser degree of slanting in the dimer cavity-lining helices in mTMEM16A compared to afTMEM16 is consistent with the lack of membrane bending upon reconstitution of the mTMEM16A channel in nanodiscs (*Dang et al., 2017*). The observed bending is similar in the $Ca^{2+}$-bound and $Ca^{2+}$-free structures presented here (*Figure 5*), consistent with the idea that the dimer cavity does not undergo $Ca^{2+}$-dependent rearrangements. In the nhTMEM16 and mTMEM16A structures, this cavity was proposed to be packed with lipids (*Brunner et al., 2014*; *Dang et al., 2017*). Indeed, while we could not identify well-defined lipids in this cavity, several densities attributable to partial acyl chains are visible between the extracellular termini of TM10 and TM2 as well as between TM3 and TM5, consistent with our hypothesis of membrane thickening around these helices (*Figure 7—figure supplement 1*). It is worth noting that reconstitution of unrelated non-scramblase membrane proteins into nanodisc, such as the SthK $K^+$ channel (*Rheinberger et al., 2018*) (*Figure 8—figure supplement 3*), do not cause distortions in the nanodisc membrane, reinforcing the idea that these effects are specific to the afTMEM16 scramblase.

The $Ca^{2+}$-dependent conformational rearrangements undergone by the pathway-lining helices correlate with changes in the density of the membrane near this conduit. In the $Ca^{2+}$-bound conformation, the pathway is open and the density between TM4 and TM6 is weaker than in the rest of the nanodisc (*Figure 5*). We propose that this weakening reflects a thinning of the membrane in this region, as the hydrophilic residues lining the open pathway provide an energetically unfavorable environment for the acyl tails. This would cause the lipids to rearrange, to allow their headgroups to interact with the polar pathway and locally thin the membrane. Indeed, similar thinning and rearrangements in lipid orientation have been proposed based on MD simulations and on the ability of afTMEM16 to scramble lipids conjugated to PEG molecules with diameters larger than the width of the lipid pathway (*Bethel and Grabe, 2016*; *Jiang et al., 2017*; *Lee et al., 2018*; *Malvezzi et al., 2018*). In the $Ca^{2+}$-free conformation, TM4 and TM6 rearrange to close the pathway preventing lipid access (*Figure 3*). The membrane exposed surface of the closed pathway is hydrophobic preventing thinning. Our finding that in the presence of the Cer24:0 lipid, the pathways of afTMEM16 are open while scrambling is inhibited suggests that its activity is not only regulated by $Ca^{2+}$-dependent gating of the protein, but that other factors, such as the physico-chemical properties of the membrane, are also critical determinants of lipid scrambling. We propose that modulation of bilayer properties and composition might constitute secondary layers of regulatory control for the in vivo activation of scramblases.

Based on these observations, we propose a mechanism for lipid scrambling where the dimer cavities 'prime' the membrane by bending the outer membrane leaflet in opposite directions at the two sides of an open lipid pathway. This creates a membrane region that is highly curved, thin and disordered, all of which will facilitate lipid transfer between leaflets through the conduit formed by the open hydrophilic pathway (*Figure 8E*) (*Bennett et al., 2009*; *Bruckner et al., 2009*; *Sapay et al., 2009*). In the $Ca^{2+}$-free conformation of the scramblase, the closed pathway prevents lipid entry and membrane thinning (*Figure 8F*). Similar mechanisms, where hydrophobic mismatches induce local distortion of membranes to lower the energy barrier for lipid movement through hydrophilic grooves, could be generally applicable to other scramblases.

## Materials and methods

**Key resources table**

| Reagent type (species) or resource | Designation | Source or reference | Identifiers | Additional information |
|---|---|---|---|---|
| Biological sample (*Saccharomyces cerevisiae*) | FYG217 -URA | doi: 10.1038/ nprot.2008.44 | | cells for expression, URA3 deletion |

*Continued on next page*

*Continued*

| Reagent type (species) or resource | Designation | Source or reference | Identifiers | Additional information |
|---|---|---|---|---|
| Biological sample (*Escherichia coli*) | BL21(DE3) | Stratagene | | Cells for MSP1E3 expression |
| Recombinant DNA reagent (synthetic gene) | MSP1E3 | Addgene https://www.addgene.org/20064/ | | |
| Recombinant DNA reagent (synthetic gene) | pET 28a | Addgene https://www.addgene.org/20064/ | | |
| Recombinant DNA reagent | pDDGFP2 (expression vector) | doi: 10.1038/nprot.2008.44 | - | |
| Recombinant DNA reagent (*Aspergillus fumigatus*) | afTMEM16 | doi:10.1038/ncomms3367 | Gene ID: 3504033 | |
| Software | Ana | M. Pusch, Istituto di Biofisica, Genova, Italy | | http://users.ge.ibf.cnr.it/pusch/programs-mik.htm |
| Software | SigmaPlot | | RRID:SCR_003210 | |
| Software | Leginon | doi: 10.1016/j.jsb.2005.03.010 | | |
| Software | Relion | doi: 10.7554/eLife.18722 | | |
| Software | MotionCorr2 | doi: 10.1038/nmeth.4193 | | |
| Software | CTFFIND4 | doi: 10.1016/j.jsb.2015.08.008 | | |
| Software | cryoSPARC | doi: 10.1038/nmeth.4169 | | |
| Software | cisTEM | doi: 10.7554/eLife.35383 | | |
| Software | Bsoft (Blocres) | doi: 10.1016/j.jsb.2006.06.006 | | |
| Software | UCSF chimera | doi: 10.1002/jcc.20084 | | |
| Software | pymol | Schrödinger | | |

## Protein expression and purification

afTMEM16 was expressed and purified as described (*Malvezzi et al., 2013*). Briefly, *S. cerevisiae* carrying pDDGFP2 (*Drew et al., 2008*) with afTMEM16 were grown in yeast synthetic drop-out medium supplemented with Uracil (CSM-URA; MP Biomedicals) and expression was induced with 2% (w/v) galactose at 30° for 22 hr. Cells were collected, snap frozen in liquid nitrogen, lysed by cry-omilling (Retsch model MM400) in liquid nitrogen (3 × 3 min, 25 Hz), and resuspended in buffer A (150 mM KCl, 10% (w/v) glycerol, 50 mM Tris-HCl, pH8) supplemented with 1 mM EDTA, 5 μg ml$^{-1}$ leupeptin, 2 μg ml$^{-1}$ pepstatin, 100 μM phenylmethane sulphonylfluoride and protease inhibitor cocktail tablets (Roche). Protein was extracted using 1% (w/v) digitonin (EMD biosciences) at 4°C for 2 hr and the lysate was cleared by centrifugation at 40,000 g for 45 min. The supernatant was supplemented with 1 mM MgCl$_2$ and 10 mM Imidazole, loaded onto a column of Ni-NTA agarose resin (Qiagen), washed with buffer A + 30 mM Imidazole and 0.12% digitonin, and eluted with buffer A + 300 mM Imidazole and 0.12% digitonin. The elution was treated with Tobacco Etch Virus protease overnight to remove the His tag and then further purified on a Superdex 200 10/300 GL column equilibrated with buffer A supplemented with 0.12% digitonin (GE Lifesciences). The afTMEM16

protein peak was collected and concentrated using a 50 $K_d$ molecular weight cut off concentrator (Amicon Ultra, Millipore).

## Liposome reconstitution and lipid scrambling assay

Liposomes were prepared as described (*Malvezzi et al., 2013*), briefly lipids in chloroform (Avanti), including 0.4% w/w tail labeled NBD-PE, were dried under $N_2$, washed with pentane and resuspended at 20 mg ml$^{-1}$ in buffer B (150 mM KCl, 50 mM HEPES pH 7.4) with 35 mM 3-[(3-cholamido-propyl)dimethylammonio]−1- propanesulfonate (CHAPS). afTMEM16 was added at 5 µg protein/mg lipids and detergent was removed using four changes of 150 mg ml$^{-1}$ Bio-Beads SM-2 (Bio-Rad) with rotation at 4°C. Calcium or EGTA were introduced using sonicate, freeze, and thaw cycles. Liposomes were extruded through a 400 nm membrane and 20 µl were added to a final volume of 2 mL of buffer B + 0.5 mM Ca(NO$_3$)$_2$ or 2 mM EGTA. The fluorescence intensity of the NBD (excitation-470 nm emission-530 nm) was monitored over time with mixing in a PTI spectrophotometer and after 100 s sodium dithionite was added at a final concentration of 40 mM. Data was collected using the FelixGX 4.1.0 software at a sampling rate of 3 Hz. All experiments with added ceramide lipids were carried out in the background of 1-palmitoyl-2-oleoyl-sn-glycero-3-phosphoethanolamine (POPE), 1-palmitoyl-2-oleoyl-sn-glycero-3-phospho-(1′-rac-glycerol) (POPG) at a ratio of 3:1. Ceramides tested include: N-stearoyl-D-erythro-sphingosine (Cer18:0), N-behenoyl-D-erythro-sphingosine (Cer22:0), N-lignoceroyl-D-erythro-sphinganine (Cer24:0), and N-nervonoyl-D-erythro-sphingosine (Cer24:1) all of which were tested at 1 mole% and 5 mole%. Chain length experiments were done in the background of 7PC:3 PG due to the availability of the long chain lipids. Lipids used include 1-palmitoyl-2-oleoyl-glycero-3-phosphocholine (POPC, 16:0-18:1C), POPG (16:0-18:1C), 1,2-dioleoyl-sn-glycero-3-phosphocholine (DOPC, 18:1C), 1,2-dioleoyl-sn-glycero-3-phospho-(1′-rac-glycerol) (DOPG, 18:1C), 1,2-dierucoyl-sn-glycero-3-phosphocholine (DEPC, 22:1C) and 1,2- dierucoyl-phosphatidylglycerol (DEPG, 22:1C).

## Quantification of scrambling activity

Quantification of the scrambling rate constants by afTMEM16 was determined as recently described (*Lee et al., 2018*; *Malvezzi et al., 2018*). Briefly, the fluorescence time course was fit to the following equation

$$F_{tot}(t) = f_0\left(L_i^{PF} + \left(1 - L_i^{PF}\right)e^{-\gamma t}\right) + \frac{(1-f_0)}{D(\alpha+\beta)}\left\{\alpha(\lambda_2 + \gamma)(\lambda_1 + \alpha + \beta)e^{\lambda_1 t} + \lambda_1\beta(\lambda_2 + \alpha + \beta + \gamma)e^{\lambda_2 t}\right\} \quad (1)$$

where $F_{tot}(t)$ is the total fluorescence at time t, $L_i^{PF}$ is the fraction of NBD-labeled lipids in the inner leaflet of protein free liposomes, $\gamma = \gamma'[D]$ where $\gamma'$ is the second order rate constant of dithionite reduction, [D] is the dithionite concentration, $f_0$ is the fraction of protein-free liposomes in the sample, $\alpha$ and $\beta$ are the forward and backward scrambling rate constants, respectively, and

$$\lambda_1 = -\frac{(\alpha+\beta+\gamma) - \sqrt{(\alpha+\beta+\gamma)^2 - 4\alpha\gamma}}{2} \quad \lambda_2 = -\frac{(\alpha+\beta+\gamma) + \sqrt{(\alpha+\beta+\gamma)^2 - 4\alpha\gamma}}{2} \quad (2)$$

$$D = (\lambda_1 + \alpha)(\lambda_2 + \beta + \gamma) - \alpha\beta$$

The free parameters of the fit are $f_0$, $\alpha$ and $\beta$ while $L_i^{PF}$ and $\gamma$ are experimentally determined from experiments on protein-free liposomes. In protein-free vesicles a very slow fluorescence decay is visible likely reflecting a slow leakage of dithionite into the vesicles or the spontaneous flipping of the NBD-labeled lipids. A linear fit was used to estimate the rate of this process was estimated to be L= $(5.4 \pm 1.6).10^{-5}$ s$^{-1}$ (n > 160). For WT afTMEM16 and most mutants the leak is >2 orders of magnitude smaller than the rate constant of protein-mediated scrambling and therefore is negligible. All conditions were tested side by side with a control preparation in standard conditions. In some rare cases, this control sample behaved anomalously, judged by scrambling fit parameters outside three times the standard deviation of the mean for the WT. In these cases, the whole batch of experiments was disregarded.

## MSP1E3 purification and nanodisc reconstitution

MSP1E3 was expressed and purified as described (*Ritchie et al., 2009*). Briefly, MSP1E3 in a pET vector (Addgene #20064) was transformed into the BL21-Gold (DE3) strain (Stratagene). Transformed cells were grown in LB media supplemented with Kanamycin (50 mg l$^{-1}$) to an $OD_{600}$ of 0.8 and expression was induced with 1 mM IPTG for 3 hr. Cells were harvested and resuspended in buffer C (40 mM Tris-HCl pH 78.0, 300 mM NaCl) supplemented with 1% Triton X-100, 5 µg ml$^{-1}$ leupeptin, 2 µg ml$^{-1}$ pepstatin, 100 µM phenylmethane sulphonylfluoride and protease inhibitor cocktail tablets (Roche). Cells were lysed by sonication and the lysate was cleared by centrifugation at 30,000 g for 45 min at 4° C. The lysate was incubated with Ni-NTA agarose resin for 1 hr at 4°C followed by sequential washes with: buffer C + 1% triton-100, buffer C + 50 mM sodium cholate +20 mM imidazole and buffer C + 50 mM imidazole. The protein was eluted with buffer C + 400 mM imidazole, desalted using a PD-10 desalting column (GE life science) equilibrated with buffer D (150 mM KCl, 50 mM Tris pH 8.0) supplemented with 0.5 mM EDTA. The final protein was concentrated to ~8 mg ml$^{-1}$ (~250 µM) using a 30 kDa molecular weight cut off concentrator (Amicon Ultra, Millipore), flash frozen and stored at −80°C.

Reconstitution of afTMEM16 in nanodiscs was carried out as follows, 3POPE:1POPG lipids in chloroform (Avanti) were dried under $N_2$, washed with pentane and resuspended in buffer D and 40 mM sodium cholate (Anatrace) at a final concentration of 20 mM. Molar ratios of 1:0.8:60 MSP1E3:afTMEM16:lipids were mixed at a final lipid concentration of 7 mM and incubated at room temperature for 20 min. Detergent was removed via incubation with Bio-Beads SM-2 (Bio-Rad) at room temperature with agitation for 2 hr and then overnight with fresh Bio-Beads SM2 at a concentration of 200 mg ml$^{-1}$. The reconstitution mixture was purified using a Superose6 Increase 10/300 GL column (GE Lifesciences) pre-equilibrated with buffer D plus 5 mM EDTA or 0.5 mM $CaCl_2$ and the peak corresponding to afTMEM16-containing nanodiscs was collected for cryo electron microscopy analysis.

## Electron microscopy data collection

3.5 µL of afTMEM16-containing nanodiscs (7 mg mL$^{-1}$) supplemented with 3 mM Fos-Choline-8-Fluorinated (Anatrace) was applied to a glow-discharged UltrAuFoil R1.2/1.3 300-mesh gold grid (Quantifoil) and incubated for one minute under 100% humidity at 15°C. Following incubation, grids were blotted for 2 s and plunge frozen in liquid ethane using a Vitrobot Mark IV (FEI). For the +C24:0 Ceramide/+$Ca^{2+}$and EDTA samples as well as +$Ca^{2+}$Dataset D (*Supplementary file 1*) micrographs were acquired on a Titan Krios microscope (FEI) operated at 300 kV with a K2 Summit direct electron detector (Gatan), using a slid width of 20 eV on a GIF Quantum energy filter and a Cs corrector with a calibrated pixel size of 1.0961 Å/pixel. A total dose of 62.61 e$^-$/Å$^2$ distributed over 45 frames (1.39 e$^-$/ Å$^2$/frame) was used with an exposure time of 9 s (200 ms/frame) and defocus range of −1.5 µm to −2.5 µm. For the +$Ca^{2+}$datasets B and C (*Supplementary file 1*), micrographs were acquired on a Titan Krios microscope (FEI) operated at 300 kV with a K2 Summit direct electron detector with a calibrated pixel size of 1.07325 Å/pixel. A total dose of 69.97 e$^-$/Å$^2$ distributed over 50 frames (1.39 e$^-$/ Å$^2$/frame) was used with an exposure time of 10 s (200 ms/frame) and a defocus range of −1.5 µm to −2.5 µm. For all samples, automated data collection was carried out using Leginon (*Suloway et al., 2005*).

## Image processing

For all datasets except dataset D (in 0.5 mM $Ca^{2+}$, *Supplementary file 1*), motion correction was performed using MotionCorr2 (*Zheng et al., 2017*) and contrast transfer function (CTF) estimation was performed using CTFFIND4 (*Rohou and Grigorieff, 2015*) both via Relion 2.0.3 (*Kimanius et al., 2016*). After manually picking ~2000 particles, the resulting 2D class-averages were used as templates for automated particle picking in Relion. The particles were extracted using a box size of 275 Å with 2xbinning and subjected to 2D classification ignoring CTFs until the first peak. For the $Ca^{2+}$-free and ceramide data sets, particles selected from 2D classification (245,835 from $Ca^{2+}$-free, and 185,451 from ceramide) were subjected to 3D classification using the nhTMEM16 crystal structure low-pass filtered to 40 Å as an initial model. For the 0.5 mM $CaCl_2$ sample, datasets B and C were used combined to make dataset A; particles selected from the first round of 2D classification from each dataset were combined and subjected to a second round of 2D classification and the resulting 302,092 particles were subjected to the same 3D classification procedure. Particles from

3D classes with defined structural features (100,268 CaCl$_2$, 70,535 Ca$^{2+}$-free, 90,709 + C24:0 ceramide,) were combined and re-extracted without binning and refined without enforcing symmetry using an initial model generated in CryoSPARC (*Punjani et al., 2017*).

Initial refinement of the afTMEM16 dataset in 0.5 mM CaCl$_2$ (Dataset B, C, which processed together form dataset A; *Supplementary file 1*) resulted in a map with a resolution of ~7 Å. The protein was symmetric, with the exception of the resolved portion of TM6 in each monomer (*Figure 1—figure supplement 2*). The complex was, however, not two-fold symmetric due to the off-center placement of the protein within the nanodisc (*Figure 1—figure supplement 2*). Therefore, data processing was carried out in parallel with C2 symmetry and without enforcing symmetry (in C1 symmetry). The particles from first C1 refinement were subjected to an additional round of 3D classification, using a mask that excluded the nanodisc and maintaining the particle orientations determined by the previous refinement. The best class from 3D classification with 27,948 particles was selected for further refinement and particle polishing. Masked refinement following particle polishing resulted in a 4.36 Å final map. To refine with C2 symmetry, the particles were polished and the nanodisc was removed using signal subtraction in Relion and the subtracted particles were refined using C2 symmetry, resulting in a 4.5 Å map. Using this map, a similar procedure to the C1 processing was carried out in which the best two classes from 3D classification without alignment applying a mask including the protein (37,146 particles) were selected for further refinement. Masked refinement of these classes yielded a 4.05 Å final density map. The C1 and C2 density maps were extensively compared and determined to be nearly identical except for the resolved portion of TM6 (*Figure 1—figure supplement 2*). The C2 map was used for model building while the C1 map was used for analysis of the afTMEM16/nanodisc complex.

For the Ca$^{2+}$-free data set (referred to as '0 Ca$^{2+}$"), the first refinement resulted in a map with a resolution of ~6 Å. As with the +Ca$^{2+}$ sample, the protein was two-fold symmetric with the exception of the resolved portion of TM6 and the overall afTMEM16/nanodisc complex was not symmetric (*Figure 1—figure supplement 2*). Therefore, data was processed in parallel using both C1 and C2 symmetries as described above. The C1 map was classified and the best class from 3D classification with a mask excluding the nanodisc (38,550 particles) was selected for further refinement and particle polishing. Masked refinement following particle polishing resulted in a 4.00 Å final density map. Masked, C2 refinement following polishing and signal subtraction resulted in a 3.89 Å map. The C2 map was used for model building while the C1 map was used for analysis of the afTMEM16/nanodisc complex in 0 Ca$^{2+}$.

For the data set in the presence of 0.5 mM CaCl$_2$ and 5 mole% C24:0 Ceramide the selected particles were refined without symmetry, which resulted in a 4.2 Å resolution map. These particles were further classified in 3D with applied mask excluding the nanodiscs and maintaining angular information from the previous 3D refinement, from which two classes with 45,021 particles were selected for masked refinement which generated a final map of 3.74 Å. The resulting refinement showed that the nanodisc and the protein were C2 symmetric, therefore, further processing was completed with C2 symmetry enforced. Refinement resulted in a map with a resolution of 3.89 Å. An additional round of 3D classification was carried out using a mask excluding the nanodisc and maintaining the particle orientations determined by the previous refinement. The highest resolution class with 24,602 particles was selected for further refinement and particle polishing. Masked refinement following particle polishing resulted in a final map with a final average resolution of 3.59 Å.

The final resolution of all maps was determined by applying a soft mask around the protein and the gold-standard Fourier shell correlation (FSC) = 0.143 criterion using Relion Post Processing (*Figure 1—figure supplement 1G*). BlocRes from the Bsoft program was used to estimate the local resolution for all final maps (*Heymann, 2001*; *Cardone et al., 2013*)(*Figure 1—figure supplement 1F*).

The two other +Ca$^{2+}$datasets (B and D, *Figure 5—figure supplement 2*, *Supplementary file 1*) mentioned were processed as described above up to the first refinement step due to the limited resolution (*Figure 5—figure supplement 2*). For processing +Ca$^{2+}$dataset D in cryoSPARC, particles were picked using Difference of Gaussian picking (*Voss et al., 2009*) implemented through the appion program suite (*Lander et al., 2009*). For analysis of the final +Ca$^{2+}$ dataset (A) in cyroSPARC, the particles originally picked in Relion were used. Extracted particles were classified in 2D and classes with structural features consistent with afTMEM16-nanodisc complexes were selected for ab initio model generation and classification. For each dataset, the best model and the associated particles were selected for homogeneous refinement followed by b-factor sharpening. Only the

C24:0 dataset had a resolution better than 6 Å from cryoSPARC so these models were not used for analysis other than to confirm the observed membrane bending and thinning (*Supplementary file 1*, *Figure 5—figure supplement 3*). For processing in cisTEM, particles from 3D classes selected in Relion were used to create an mrc stack and imported. Several rounds of automatic refinement and manual refinement with both local and global searches were carried out. Using auto-masking and manual refinement the final resolution for the +Ca$^{2+}$dataset was~5 Å and was therefore not used for an analysis other than to confirm the observed effects on the membrane (*Supplementary file 1*, *Figure 5—figure supplement 3*).

## Model building, refinement, and validation

The maps were of high quality and we built atomic models for each structure for the transmembrane domain, most of the cytosolic region and connecting loops (*Figure 1—figure supplement 4*), as well as identifying seven partial lipids in the +C24:0 Ceramide structure (*Figure 7—figure supplement 1*). The model of afTMEM16 in the presence of ceramide was built first, due to the higher resolution of the map. The crystal structure of nhTMEM16 (PDBID 4wis) was used as a starting model and docked into the density map using chimera and the jiggle fit script in COOT (*Emsley et al., 2010*) (https://www2.mrc-lmb.cam.ac.uk/groups/murshudov/index.html), and mutated to match the afTMEM16 sequence. The final model contains residues 13–54, 60–119, 126–259, 270–312, 316–399, 418–461, 490–597, 602–657 and 705–724 and the following residues were truncated due to missing side chain density: L43, Q49, R53, K69, K70, E94, K100, Q102, K129, H130, D137, K257, E258, L316, E461, H555, F705, K713, E714, and E717. The model was improved iteratively by real space refinement in PHENIX imposing crystallographic symmetry and secondary structure restraints followed by manual inspection and removal of outliers. The model for the +Ca$^{2+}$structure was built using the refined +C24:0 Ceramide structure (PDBID 6E1O) as an initial model, that was docked into the +Ca$^{2+}$density using the jigglefit script in COOT (*Emsley et al., 2010*) (https://www2.mrc-lmb.cam.ac.uk/groups/murshudov/index.html), manually inspected, and refined using real space refinement in PHENIX as above (*Adams et al., 2010*; *Afonine et al., 2013*). The final model contains residues 13–54, 60–119, 128–258, 270–312, 316–400, 420–460, 490–594, 602–659 and 705–724, and the following residues were truncated due to missing side chain density: D17, Q49, R53, K100, Q102, E104, K129, H130, N135, D137, E164, H247, K254, K257, H460, K550, H595, R710, K713, E714, E717, and L720. The model for the 0 Ca$^{2+}$ structure was built using the +Cera model as a starting point and regions where the density differed were built manually (TM3, TM4 and TM6) or rigid body refined where smaller rearrangements were observed. The final model contains residues 13–54, 60–119, 128–258, 270–312, 316–400, 420–460, 490–594, 602–659 and 705–724, and the following residues were truncated due to missing side chain density: D17, F36, R53, K70, E94, K100, Q102, E132, K241, H247, R250, K257, E310, K317, K634, K642, Y654, R704, R708, E714, E717. For all three structures, the density in the regions of TM3 and TM4 and the connecting loop (residues 274–352) was less well-defined than remainder of the structure and the model may be less accurate in this area.

To validate the refinement, the FSC between the refined model and the final map was calculated (FSCsum) (*Figure 1—figure supplement 1H*). To evaluate for over-fitting, random shifts of up to 0.3 Å were introduced in the final model and the modified model was refined using PHENIX against one of the two unfiltered half maps. The FSC between this modified-refined model and the half map used in refinement (FSCwork) was determined and compared to the FSC between the modified-refined model and the other half map (FSCfree) which was not used in validation or refinement. The similarity in these curves indicates that the model was not over-fit (*Figure 1—figure supplement 1H*). The quality of all three models was assessed using MolProbity (*Chen et al., 2010*) and EMRinger (*Barad et al., 2015*), both of which indicate that the models are of high quality (*Figure 1—figure supplement 1H*).

## Difference map calculation

To compare the maps resulting from C1 and C2 processing in the 0 Ca$^{2+}$ and +Ca$^{2+}$ structures, we calculated a difference map between the two volumes using the volume operation subtract function in chimera (*Pettersen et al., 2004*) (*Figure 1—figure supplement 2*). We used 'omit density' to assign the placement of several lipids in the 0 Ca$^{2+}$ and +C24:0 Ceramide structures and the Ca$^{2+}$ ions in the +Ca$^{2+}$and + Cera structures which were calculated using the 'phenix.

real_space_diff_map' function in PHENIX (*Adams et al., 2010*; *Afonine et al., 2013*; *Dang et al., 2017*). Briefly, completed models without the ligand in question were used to generate a theoretical density map which was subtracted from the experimental density map. In the subtracted map, areas of ligand density appeared as positive density.

## Gramicidin fluorescence assay

The gramicidin channel-based fluorescence assay for monitoring changes in lipid bilayer properties was implemented using large unilamellar vesicles (LUVs) incorporating gramicidin (gA) and loaded with 8-aminonaphthalene-1,3,6-trisulfonic acid, disodium salt (ANTS), following published protocols (*Ingólfsson and Andersen, 2010*; *Ingólfsson et al., 2011*). Vesicles composed of 1-palmitoyl-2-oleoyl-sn-glycero-3-phosphoethanolamine (POPE) and 1-palmitoyl-2-oleoyl-sn-glycero-3-phospho-1'-rac-glycerol (POPG) in 3:1 (mol/mol) proportion in chloroform were mixed with a gA to lipid molar ratio of 1:40,000. An additional 0, 1, or 5 mole% of C24:0 ceramide (d18:1/24:0) or C24:1 ceramide (d18:1/24:1) in chloroform were added to the gA-phospholipid mix. The lipid mixtures were dried under nitrogen to remove chloroform and further dried in vacuum overnight. Lipids were rehydrated with fluorophore-containing buffer solution (25 mM ANTS, 100 mM $NaNO_3$, and 10 mM HEPES), vortexed for 1 min, and incubated at room temperature for at least 3 hr while shielded from light. The solutions were sonicated for 1 min at low power and subjected to six freeze-thaw cycles. The samples then were extruded 21 times with an Avanti Polar Lipids mini-extruder (Avanti) and 0.1 mm polycarbonate filter. The samples were extruded at 35–40°C. Extravesicular ANTS was removed with PD-10 desalting column (GE Healthcare). The average LUV diameter was ~130 nm, with an average polydispersity index (PDI) of 10%, as determined using dynamic light scattering. LUV stock solutions were stored at room temperature with an average shelf life of 7 days.

The time course of the ANTS fluorescence was measured at 25°C with an SX-20 stopped-flow spectrofluorometer (Applied Photophysics), with an LED light source. Excitation was at 352 nm and emission was recorded above 450 nm with a high-pass filter; the deadtime of the instrument is $\approx$ 2 ms with a sampling rate of 5000 points/s. Samples were prepared by diluting the stock lipid concentration with buffer solution (140 mM NaOH and 10 mM HEPES) to 125 µM LUV; each sample was incubated for at least 10 min before several 1 s mixing reactions. Each sample was first mixed with the control buffer (no $Tl^+$), followed by mixing with the quench solution (50 mM $TlNO_3$ 94 mM $NaNO_3$ and 10 mM HEPES). Experiments are conducted with two independently prepared populations of vesicles per lipid/ceramide combination and traces are analyzed in MATLAB (MathWorks).

The fluorescence quench rate therefore was determined as described (*Ingólfsson and Andersen, 2010*) by fitting the time course to a stretched exponential function (*Berberan-Santos et al., 2005*):

$$F(t) = F(\infty) + [F(0) - F(\infty)] \cdot \exp\left[-(t/\tau_0)^\beta\right] \tag{3}$$

*F(t)* denotes the fluorescence intensities at time *t*; β (0 < β ≤1) is a parameter that accounts for the dispersity of the vesicle population; and $\tau_o$ is a parameter with dimension of time. *F(0)*, *F(∞)*, β and $\tau_o$ were determined from a nonlinear least squares fit of *Eq. 1* to each individual fluorescence trace, and the quench rate was determined (*Berberan-Santos et al., 2005*):

$$Rate(t) = \frac{\beta}{\tau 0}\left[\frac{t}{\tau 0}\right]^{(\beta-1)} \tag{4}$$

evaluated at *t* = 2 ms. The ceramide-induced changes in the quench rate then was evaluated as

$$Rate/Rate_{cntrl} = \frac{Rate(t)_{ceramide}}{Rate(t)_{cntrl}} \tag{5}$$

where the subscripts 'ceramide' and 'cntrl' denote the rates observed in the presence and absence of ceramide.

## Data availability

The three-dimensional cryo-EM density maps of the calcium-bound, calcium-free, and $Ca^{2+}$-bound in the presence of C24:0 Ceramide afTMEM16/nanodisc complexes have been deposited in the Electron Microscopy Data Bank under accession numbers EMD-8948, EMD-8931, and EMDB-8959,

respectively. The deposition includes corresponding masked and unmasked maps, the mask used for the final FSC calculation, and the FSC curves. Coordinates for the models of the calcium-bound, calcium-free, and $Ca^{2+}$-bound ceramide inhibited states have been deposited in the Protein Data Bank under accession numbers 6E0H, 6DZ7, and 6E1O respectively. All other data are available from the corresponding author upon reasonable request.

## Acknowledgements

The authors thank members of the Accardi lab for helpful discussions. Richard Hite for helpful suggestions on cryo-EM data processing, Christopher Miller and Simon Scheuring for insightful discussions on the work, Jeremy Dittman for insightful discussions and artistic contributions to the work, Eva Fortea Verdejo for help with sequence alignments and artistic contributions, Olga Boudker and Xiaoyu Wang for help with the nanodisc preparation. This work was supported by NIH Grants R01GM106717 (to AA), 1R01GM124451-02 (to CMN) and R21NS10451 (to ADL), an Irma T Hirschl/ Monique Weill-Caulier Scholar Award (to AA), by the Basic Science Research Program through the National Research Foundation of Korea (NRF) funded by the Ministry of Education, Science and Technology (grant 2013R1A6A3A03064407 to B-C L) and by KBRI basic research program through Korea Brain Research Institute funded by Ministry of Science and ICT (18-BR-01–02 to B-CL). MEF is the recipient of a Weill Cornell Medicine Margaret and Herman Sokol Fellowship. All EM data collection and screening was performed at the Simons Electron Microscopy Center and National Resource for Automated Molecular Microscopy located at the New York Structural Biology Center, supported by grants from the Simons Foundation (349247), NYSTAR, and the NIH National Institute of General Medical Sciences (GM103310) with additional support from Agouron Institute (F00316) and NIH S10 OD019994-01. Initial negative stain screening was performed at the Weill Cornell Microscopy and Image Analysis Core Facility, with the help of L Cohen-Gould, and at the Rockefeller University Electron Microscopy Resource Center, with the help of Kunihiro Uryu and Devrim Acehan.

## Additional information

### Funding

| Funder | Grant reference number | Author |
|---|---|---|
| National Institute of General Medical Sciences | R01GM106717 | Alessio Accardi |
| Irma T. Hirschl Trust | | Alessio Accardi |
| Margaret and Herman Sokol Fellowship | | Maria E Falzone |
| National Research Foundation of Korea | 2013R1A6A3A03064407 | Byoung-Cheol Lee |
| Agouron Institute | F00316 | Edward T Eng |
| Simons Foundation | 349247 | Edward T Eng |
| National Institute of General Medical Sciences | GM103310 | Edward T Eng |
| National Institute of General Medical Sciences | 1R01GM124451-02 | Crina M Nimigean |
| Ministry of Science, ICT and Future Planning | 18-BR-01-02 | Byoung-Cheol Lee |
| National Institute of Neurological Disorders and Stroke | R21NS10451 | Annarita Di Lorenzo |

The funders had no role in study design, data collection and interpretation, or the decision to submit the work for publication.

## Author contributions
Maria E Falzone, Conceptualization, Data curation, Formal analysis, Supervision, Validation, Investigation, Visualization, Methodology, Writing—original draft, Writing—review and editing; Jan Rheinberger, Formal analysis, Methodology, Writing—review and editing; Byoung-Cheol Lee, Linda Sasset, Ashleigh M Raczkowski, Annarita Di Lorenzo, Investigation; Thasin Peyear, Edward T Eng, Olaf S Andersen, Investigation, Writing—review and editing; Crina M Nimigean, Funding acquisition, Methodology, Writing—review and editing; Alessio Accardi, Conceptualization, Data curation, Formal analysis, Supervision, Funding acquisition, Validation, Visualization, Methodology, Writing—original draft, Project administration, Writing—review and editing

## Author ORCIDs
Maria E Falzone (iD) http://orcid.org/0000-0001-6738-7017
Jan Rheinberger (iD) http://orcid.org/0000-0002-9901-2065
Edward T Eng (iD) http://orcid.org/0000-0002-8014-7269
Crina M Nimigean (iD) http://orcid.org/0000-0002-6254-4447
Alessio Accardi (iD) http://orcid.org/0000-0002-6584-0102

## Decision letter and Author response
Decision letter https://doi.org/10.7554/eLife.43229.057
Author response https://doi.org/10.7554/eLife.43229.058

# Additional files

## Supplementary files
• Source data 1. Raw data of the representative fluorescence decay traces of afTMEM16-mediated lipid scrambling in liposomes formed from lipids with different chain length and saturation.
DOI: https://doi.org/10.7554/eLife.43229.032

• Source data 2. Raw data of the representative fluorescence decay traces of afTMEM16-mediated lipid scrambling in liposomes containing different Ceramide lipids.
DOI: https://doi.org/10.7554/eLife.43229.033

• Supplementary file 1. Summary of cryo-EM datasets utilized in this work. Detailed processing procedures are described in the methods. Note that +$Ca^{2+}$dataset C was analyzed independently only up to 2D classification; after 2D classification it was combined with dataset B to generate dataset A, which yielded the final high-resolution +$Ca^{2+}$ map.
DOI: https://doi.org/10.7554/eLife.43229.035

• Supplementary file 2. Average values of the scrambling rate constants of afTMEM16 in short (16–18C) and long (22:1) chain lipids. The following parameters were derived as described [6] by fitting the data to *Eq. 1*: $f_0$ is the fraction of empty liposomes, $\alpha$ and $\beta$ are the forward and reverse scrambling rate constants, $\gamma$ is the reduction rate constant by dithionite, $L_i^{PF}$ is the fraction of NBD-labeled lipids in the inner leaflet of a protein-free vesicle, n is the number of independent experiments. Data is reported as the mean ±SD. * denotes values that were constrained during fitting.
DOI: https://doi.org/10.7554/eLife.43229.036

• Supplementary file 3. Statistics of cryo-EM data collection, 3D reconstruction and model refinement.
DOI: https://doi.org/10.7554/eLife.43229.034

• Supplementary file 4. Average values of the scrambling rate constants of afTMEM16 in the presence of various ceramides. The following parameters were derived by fitting the data to *Eq. 1*: $f_0$ is the fraction of empty liposomes, $\alpha$ and $\beta$ are the forward and backward scrambling rate constants, $\gamma$ is the reduction rate constant by dithionite, $L_i^{PF}$ is the fraction of NBD-labeled lipids in the inner leaflet of a protein-free vesicle, n is the number of independent experiments. Data is reported as the mean ±SD.
DOI: https://doi.org/10.7554/eLife.43229.037

## Data availability

Masked and unmasked cryoEM maps have been deposited in the EMDB database under accession codes: EMD-8931, EMD-8948, EMD-8959. Atomic coordinates have been deposited in the PDB database under accession codes: 6E0H, 6DZ7, 6E1O. Motion corrected EM micrographs are deposited in the EMPIAR database under accession codes: EMPIAR-10240, EMPIAR-10239,EMPIAR-10241. Key parameters of the fluorescence time courses are detailed in the tables and figures and representative traces used in the figures have been provided as source data files. The raw fluorescence time courses are available upon request to the corresponding author.

The following datasets were generated:

| Author(s) | Year | Dataset title | Dataset URL | Database and Identifier |
|---|---|---|---|---|
| Falzone ME, Accardi A | 2018 | afTMEM16 reconstituted in nanodiscs in the presence of Ca2+ | https://www.rcsb.org/structure/6E0H | Protein Data Bank, 6E0H |
| Falzone ME, Accardi A | 2018 | CryoEM data from Structural basis of Ca2+-dependent activation and lipid transport by a TMEM16 scramblase | https://www.rcsb.org/structure/6D27 | Protein Data Bank, 6D27 |
| Falzone ME, Accardi A | 2018 | afTMEM16 reconstituted in nanodiscs in the presence of Ca2+ and ceramide 24:0 | https://www.rcsb.org/structure/6E1O | Protein Data Bank, 6E1O |
| Falzone ME, Accardi A | 2018 | Ca2+-free afTMEM16 in nanodisc | http://www.ebi.ac.uk/pdbe/entry/emdb/EMD-8931 | Electron Microscopy Data Bank, EMD-8931 |
| Falzone ME, Accardi A | 2018 | Ca2+-bound afTMEM16 in nanodisc | http://www.ebi.ac.uk/pdbe/entry/emdb/EMD-8948 | Electron Microscopy Data Bank, EMD-8948 |
| Falzone ME, Accardi A | 2018 | Structure of afTMEM16 in nanodisc in the presence of 0.5 mM Ca2+ and 0.5 mol% Ceramide 24:0 | http://www.ebi.ac.uk/pdbe/entry/emdb/EMD-8959 | Electron Microscopy Data Bank, EMD-8959 |
| Maria E Falzone, Alessio Accardi | 2019 | Motion Corrected micrographs of the afTMEM16/nanodisc complex in the absence of Ca2+ | https://www.ebi.ac.uk/pdbe/emdb/empiar/entry/10241/ | EMPIAR, EMPIAR-10241 |
| Maria E Falzone, Alessio Accardi | 2019 | Motion Corrected micrographs of the afTMEM16/nanodisc complex in the presence of 0.5 mM Ca2+ | https://www.ebi.ac.uk/pdbe/emdb/empiar/entry/10239/ | EMPIAR, EMPIAR-10239 |
| Maria E Falzone, Alessio Accardi | 2019 | Motion Corrected micrographs of the afTMEM16/nanodisc complex in the presence of 0.5 mM Ca2+ and 5 mol% Ceramide 24:0 | https://www.ebi.ac.uk/pdbe/emdb/empiar/entry/10240/ | EMPIAR, EMPIAR-10240 |

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
