## [Decision Letter]

Thank you for submitting your article "Cryo-EM structures reveal bilayer remodeling during Ca^2+^ activation of a TMEM16 scramblase" for consideration by *eLife*. Your article has been reviewed by three peer reviewers, including X as the Reviewing Editor and Reviewer #1, and the evaluation has been overseen by a Reviewing Editor and Richard Aldrich as the Senior Editor. The following individuals involved in review of your submission have agreed to reveal their identity: Stephen Barstow Long (Reviewer #3).

The reviewers have discussed the reviews with one another and the Reviewing Editor has drafted this decision to help you prepare a revised submission.

Summary:

This interesting manuscript presents three nice cryo-EM structures of afTMEM16 in nanodiscs. The cryo-EM is well-done. The densities support the atomic models and are consistent with FSC-indicated resolution limits. The three structures are a significant advance in our understanding of TMEM16 scramblases. First, they represent the first structures of TMEM16 scramblases determined in lipid nanodiscs (a previous structure of nhTMEM16 scramblase had been determined in detergent). Second, they are determined in the presence and absence of Ca^2+^. The structures thereby reveal Ca^2+^-dependent conformational changes that are mechanistically satisfying, consistent with existing functional and mutational data, and allow the authors to propose a plausible model for lipid scrambling by this class of enzymes. There are two novel conclusions: first, gating of the afTMEM16 scramblase by Ca^2+^ involves global conformational changes that affect the organization of the lipid bilayer and, second, that scrambling is dependent on the species of lipids present in the reconstitution. In terms of presentation, these are two independent stories that could be better integrated and focused. The first story about TMEM16 gating is solid, but the second story is not as well developed.

Essential revisions:

1) The major focus of the manuscript should be on the remarkable conformational changes in the protein that are induced by the binding of Ca^2+^ and the lipid-scrambling pathway that becomes present only when Ca^2+^ is bound. The authors certainly make these points, but they could be strengthened and clarified. They certainly warrant a movie showing the transition and the formation of the lipid pathway when Ca^2+^ is bound. Furthermore, we found the manuscript somewhat confusing and a bit disorganized at times. A series of suggestions below (see point 4) should help clarify these issues.

2) We feel uneasy about the emphasis placed upon what the authors refer to as "bilayer remodeling". The structures are determined in lipid nanodiscs. Therefore, the density surrounding the transmembrane region of afTMEM16 would be due to combination of the nanodisc scaffold protein and lipid molecules. The authors observe that the TMEM16 protein is off-center in the nanodisc in the datasets in which Ca^2+^ was included. This off-centered nature is an interesting point, and we agree that it potentially provides a clue to the mechanism, but we would not be comfortable saying that this is direct evidence that the protein causes bilayer remodeling in the presence of Ca^2+^. For example, is it possible that the classification of the particles identified those that were off-center? Perhaps the large size of the nanodisc allows the TMEM16 protein to "rattle around" within it such that its distribution within the nanodisc is perturbed by minor effects. The authors state (subsection “Ca^2+^-dependent remodeling of the nanodisc upon scramblase activation”): "We… observed clear, Ca^2+^-dependent remodeling of the membrane around the protein in both the 2D class averages and 3D reconstructions (Figure 5A-D; Figure 5—figure supplement 1)." We could not identify in the 2D class averages what the authors were referring to: both 0 Ca^2+^ and 0.5 mM Ca^2+^ seemed to show asymmetry within the nanodisc. Thus, it is unclear what is meant by "nanodiscs containing afTMEM16 are more distorted in the presence than in the absence of Ca^2+^". To our mind, there are many possible explanations for the asymmetry, and "bilayer remodeling" should not be a major focus of the manuscript or its Title. Perhaps the emphasis could be shifted to the Ca^2+^-dependent conformational changes that are observed in a lipid environment.

3) The data on the effects of lipid chain length and ceramide require additional work. The lipid effects are more complicated than appears at first glance. The experiments on the effects on chain length are weakened by the limited variety of lipids tested and the fact that the C22 lipids not only differ in chain length but also in saturation. Figure 6B was hard to decipher because the denotation 3PE:1PG and 7PG:3PG made it seem that both mixtures contained saturated acyl chains, but Methods indicates that the lipids in C22 PC:PG (DEPC and DEPG) contain two unsaturated fatty acids, while the lipids in C16-18 PC:PG (POPC and POPG) contain only one. Thus, we do not know whether the inhibition of scrambling in Figure 6B is due to chain length or saturation. Is this inhibition actually related to the membrane thickness as the authors suppose or does DEPC or DEPG inhibit afTMEM16 directly? What is the effect of addition of 5% DEPC or DEPG to POPE:POPG liposomes? The observation that C24:1 ceramide has no effect is fascinating, but does C22:1 also have no effect? Also, ceramide not only affects Ca^2+^-dependent scrambling, but also inhibits Ca^2+^-independent scrambling (especially true for C22:0, C24:0, and C24:1). The scrambling with 5% C24:0 ceramide in the absence of protein seems to be significantly augmented as if the liposome integrity has been compromised.

We see two ways of addressing these concerns. The authors might solidify the suggested mechanism by showing (1) scrambling rates in PC:PG with C22 saturated acyl chains, (2) the effect of 5% DEPC/DEPG in POPE:POPG liposomes, and (3) the effect of C22:1 ceramide. We realize that some of these experiments may be constrained by the commercial availability of lipids. Thus, if saturated C22 PC:PG is not available commercially, the authors could drop (1), but show the concentration dependence of DEPC/DEPG in a POPE:POPG background and try to relate this to membrane thickness (although formation lipid microdomains might pose a complication). Alternatively, the authors could remove most of the functional data, and just show inhibition by ceramide and acknowledge that more work will need to be done to clarify its mechanism. The cryo-EM structure of nhTMEM16 with ceramide is intriguing, but unfortunately does not provide any mechanistic insight into ceramide inhibition of the scrambling.

4) Suggestions for improving presentation clarity:

4.1) To benefit the reader who is unfamiliar with the other structures of TMEM16 proteins, it would be beneficial to provide additional context to the current structures. E.g.: They represent the first structures of TMEM16 scramblases determined in lipid nanodiscs – a previous structure of nhTMEM16 scramblase had been determined in detergent. They also represent first structural information for afTMEM16. It could be emphasized that a lipid environment is particularly important to understand the conformational changes +/- Ca^2+^ (e.g.: Ca^2+^ is found within the TM region, the enzyme may need its lipid substrates to adopt the proper conformation, etc.)

4.2) The first subsection is titled "Structure of the Ca^2+^-bound afTMEM16 scramblase in nanodisc", but the section goes on to discuss both the Ca^2+^-bound structure and conformational changes in the absence of Ca^2+^. Moreover, the first paragraph of this section is a bit confusing because it was difficult to discern the important information. Terms were introduced without context (e.g. the "hydrophobic dimer cavity"). The discussion of C1 vs. C2 might be better placed elsewhere.

4.3) In subsection "Ca^2+^ binding induces global rearrangements in afTMEM16" Ca^2+^-induced conformational changes are described partly as a movement going from the Ca^2+^-free to the Ca^2+^-bound state (sentences starting with "The cytosolic domains of afTMEM16 are translated…" and "In the absence of Ca^2+^…"), but partly as a movement going from the Ca^2+^-bound to the Ca^2+^-free state (sentence starting with "From the Ca^2+^-bound conformation…"). This is confusing to the reader. It might be better to describe conformational differences between the two structures consistently as going from one particular conformation to the other.

4.4) It is questionable whether lipids (potentially observed in the hydrophobic dimer cavity) can be resolved at 3.9Å resolution (the 0 Ca^2+^ structure). If mentioned (for the 0 Ca^2+^ or ceramide structures), it would be more accurate to refer to the "lipids" as "density consistent with the acyl chains of lipids" or something similar since densities for intact lipid molecules are not observed. The atomic coordinates of the 0 Ca^2+^ structure do not include lipids – they should either be included in the coordinates or removed altogether.

---

## [Author Response]

Essential revisions:1) The major focus of the manuscript should be on the remarkable conformational changes in the protein that are induced by the binding of Ca^2+^ and the lipid-scrambling pathway that becomes present only when Ca^2+^ is bound. The authors certainly make these points, but they could be strengthened and clarified. They certainly warrant a movie showing the transition and the formation of the lipid pathway when Ca^2+^ is bound. Furthermore, we found the manuscript somewhat confusing and a bit disorganized at times. A series of suggestions below (see point 4) should help clarify these issues.

As suggested by the reviewers, we extensively re-organized the manuscript, harmonized the presentation of the results and added emphasis to the protein conformational changes.

2) We feel uneasy about the emphasis placed upon what the authors refer to as "bilayer remodeling". The structures are determined in lipid nanodiscs. Therefore, the density surrounding the transmembrane region of afTMEM16 would be due to combination of the nanodisc scaffold protein and lipid molecules. The authors observe that the TMEM16 protein is off-center in the nanodisc in the datasets in which Ca^2+^ was included. This off-centered nature is an interesting point, and we agree that it potentially provides a clue to the mechanism, but we would not be comfortable saying that this is direct evidence that the protein causes bilayer remodeling in the presence of Ca^2+^. For example, is it possible that the classification of the particles identified those that were off-center? Perhaps the large size of the nanodisc allows the TMEM16 protein to "rattle around" within it such that its distribution within the nanodisc is perturbed by minor effects. The authors state (subsection “Ca^2+^-dependent remodeling of the nanodisc upon scramblase activation”): "We… observed clear, Ca^2+^-dependent remodeling of the membrane around the protein in both the 2D class averages and 3D reconstructions (Figure 5A-D; Figure 5—figure supplement 1)." We could not identify in the 2D class averages what the authors were referring to: both 0 Ca^2+^ and 0.5 mM Ca^2+^ seemed to show asymmetry within the nanodisc. Thus, it is unclear what is meant by "nanodiscs containing afTMEM16 are more distorted in the presence than in the absence of Ca^2+^". To our mind, there are many possible explanations for the asymmetry, and "bilayer remodeling" should not be a major focus of the manuscript or its Title. Perhaps the emphasis could be shifted to the Ca^2+^-dependent conformational changes that are observed in a lipid environment.

As suggested, we removed bilayer remodeling from the Title, overall de-emphasized remodeling and the asymmetry of the complexes, and more clearly separate the description of the data from our interpretation.

We renamed the subsection on membrane remodeling to “Structural analysis of the afTMEM16/nanodisc complex”, we significantly shortened it and moved some of the more speculative portions to the Discussion section. We apologize for not clearly explaining what we mean by membrane remodeling. We now state that “Inspection of these maps suggests the membrane in nanodiscs containing afTMEM16 is bent along the two dimer cavities of the scramblase (Figure 5A, C) and that there is a region of low density at the open lipid pathway (Figure 5B).” We added several figures to better illustrate how our data supports these statements. Briefly:

i) We see the membrane bending along both dimer cavities in 5 datasets (3 in the presence of 0.5 mM Ca^2+^, one in Ca^2+^ free and one with 0.5 mM Ca^2+^ and 5 mole% C24:0 ceramide). This is shown in Figure 5A, C and Figure 5—figure supplements 1, 2 and 3.

ii) We see weakening of the density at both lipid pathways in the 3 datasets in the presence of Ca^2+^ (when the pathway is open). This is shown in Figure 5B, and Figure 5—figure supplements 1 and 2.

iii) These effects are not dependent on the data processing algorithm. We show that re-processing the high-resolution dataset in 0.5 mM Ca^2+^ with Relion, cryoSPARC or cisTEM yields similar looking maps of the afTMEM16/nanodisc complex. This is shown in Figure 5—figure supplement 3. The different processing strategies for all datasets are reported in Supplementary file 2.

iv) These effects are specific to the afTMEM16 scramblase. Reconstitution of other membrane proteins in nanodiscs is not generally associated to membrane distortions. We show the low pass filtered unmasked map of the SthK K^+^ channel in nanodiscs (Rheinberger et al., 2018) in Figure 8—figure supplement 3 as an example and note that the channel-only homologue TMEM16A does not distort nanodiscs (Dang et al., 2017). This is now discussed in the Discussion section.

We note that a recent report from the Dutzler and Paulino labs describes similar findings for the nhTMEM16. They report bending of the nanodisc membrane and of the detergent micelle at the dimer cavities similar to what we see. This is reported in a manuscript currently available online in bioRxiv (see Figure 5 in Kalienkova et al., 2018).

The reviewers raise 3 main technical concerns:

i) “The structures are determined in lipid nanodiscs. Therefore, the density surrounding the transmembrane region of afTMEM16 would be due to combination of the nanodisc scaffold protein and lipid molecules.”

We agree with the reviewers that the nanodisc density is the combination of the nanodisc and MSP1E3 protein and that it is likely that the afTMEM16 scramblase distorts both to a similar extent. This is now noted in subsection “Structural analysis of the afTMEM16/nanodisc complex”.

However, we do not think that the effects we describe are due to the scaffold protein, which is localized at and defines the outer edge of the nanodisc. If the MSP protein contributed to the membrane density, then we would expect that its effect should depend on the relative position of the protein within the nanodisc. In contrast, we observe bending at both dimer cavities in 5 datasets, regardless of the positioning of the protein within the nanodisc (Figure 5—figure supplements 1 and 2). Similarly, the weakening of the density is seen at both open pathways of the 3 datasets obtained in the presence of 0.5 mM Ca^2+^ (Figure 5B, Figure 5—figure supplement 2). Thus, the same effects are seen regardless of the distance from the nanodisc edge suggesting that the density of the MSP does not contribute significantly to them.

ii) “The authors observe that the TMEM16 protein is off-center in the nanodisc in the datasets in which Ca^2+^ was included. This off-centered nature is an interesting point, and we agree that it potentially provides a clue to the mechanism, but we would not be comfortable saying that this is direct evidence that the protein causes bilayer remodeling in the presence of Ca^2+^.”

We agree that while the acentric position of the protein within the nanodisc in the presence of Ca^2+^ is interesting, we cannot draw mechanistic insights from it. To minimize confusion, we describe this observation only at the beginning of the manuscript, where we discuss the processing of the data in C1 vs. C2 symmetry (Figure 1—figure supplements 1 and 2).

iii) “We could not identify in the 2D class averages what the authors were referring to: both 0 Ca^2+^ and 0.5 mM Ca^2+^ seemed to show asymmetry within the nanodisc.”

We thank the reviewers for pointing this out. We now clarify that the 2D classes show that the nanodisc is bent both in the presence and absence of Ca^2+^. This is consistent with the observation that the membrane bending along the dimer cavities is seen independently of Ca^2+^.

3) The data on the effects of lipid chain length and ceramide require additional work. The lipid effects are more complicated than appears at first glance. The experiments on the effects on chain length are weakened by the limited variety of lipids tested and the fact that the C22 lipids not only differ in chain length but also in saturation. Figure 6B was hard to decipher because the denotation 3PE:1PG and 7PG:3PG made it seem that both mixtures contained saturated acyl chains, but Methods indicates that the lipids in C22 PC:PG (DEPC and DEPG) contain two unsaturated fatty acids, while the lipids in C16-18 PC:PG (POPC and POPG) contain only one. Thus, we do not know whether the inhibition of scrambling in Figure 6B is due to chain length or saturation. Is this inhibition actually related to the membrane thickness as the authors suppose or does DEPC or DEPG inhibit afTMEM16 directly?

We thank the reviewers for raising this excellent point. We carried out additional experiments where we form liposomes using a 7:3 DOPC:DOPG mixture. These lipids have two chains with 18 carbons and with one unsaturated bond, therefore directly allow us to test the effect of saturation on scrambling. Our results show that afTMEM16 is fully functional in these lipids, indicating that presence of two saturated tails does not inhibit scrambling. These data have been added to Figure 6 and Figure 6—figure supplement 1 and are discussed in subsection “Lipid dependence of scrambling by afTMEM16”.

What is the effect of addition of 5% DEPC or DEPG to POPE:POPG liposomes? The observation that C24:1 ceramide has no effect is fascinating, but does C22:1 also have no effect?

Unfortunately, the C22:1 ceramide lipid is not commercially available preventing us from experimentally testing its effect.

Also, ceramide not only affects Ca^2+^-dependent scrambling, but also inhibits Ca^2+^-independent scrambling (especially true for C22:0, C24:0, and C24:1).

The effect of the long chain ceramides on scrambling in 0 Ca^2+^ is now discussed

subsection “Lipid dependence of scrambling by afTMEM16”.

The scrambling with 5% C24:0 ceramide in the absence of protein seems to be significantly augmented as if the liposome integrity has been compromised.

Addition of 5% C24:0 leads to an asymmetric initial distribution of the NBD-PE probe, as shown by the fact that the steady state value of the fluorescence decay of the protein-free liposomes dips below the 50% mark. This asymmetry does not reflect compromised liposome integrity, as evidenced by the fact that the fluorescence reaches a plateau and remains stable for the duration of the experiment. In the case of leaky liposomes we observe a slow decay of fluorescence towards 0 as all fluorophores become bleached. This is now noted in the figure legend to Figure 6—figure supplement 2. We observe this asymmetric distribution in a variety of different lipid compositions (see for example Malvezzi et al., 2013, or Malvezzi et al., 2108) without adverse effects on liposome integrity. Indeed, a similar asymmetric initial distribution is visible in the long chain DEPC:DEPG liposomes (Figure 6—figure supplement 1D). While do not know the origins of this preferential direction of incorporation of the lipids, we speculate it is likely due to headgroup packing and membrane curvature. Importantly, this asymmetry does not affect our estimate of the effects of the C24:0 on the scrambling rate constants. In fitting the time course of fluorescence decay in proteoliposomes to Equation 1, the fraction of labeled lipids in the inner leaflet is fixed to the value measured in the protein free liposomes from a same-day reconstitution. Please see Malvezzi et al. (2018) for a complete description of this analysis.

We see two ways of addressing these concerns. The authors might solidify the suggested mechanism by showing (1) scrambling rates in PC:PG with C22 saturated acyl chains, (2) the effect of 5% DEPC/DEPG in POPE:POPG liposomes, and (3) the effect of C22:1 ceramide. We realize that some of these experiments may be constrained by the commercial availability of lipids. Thus, if saturated C22 PC:PG is not available commercially, the authors could drop (1), but show the concentration dependence of DEPC/DEPG in a POPE:POPG background and try to relate this to membrane thickness (although formation lipid microdomains might pose a complication). Alternatively, the authors could remove most of the functional data, and just show inhibition by ceramide and acknowledge that more work will need to be done to clarify its mechanism.

As discussed above, we performed experiments in DOPC:DOPG to address the reviewers’ concerns and show that acyl chain saturation has no effect on lipid scrambling. This approach avoids concerns about the formation of microdomains that might arise in mixed chain length membranes.

Unfortunately, neither the C22 PG lipids with one saturated chain nor the C22:1 ceramide lipids are commercially available, preventing us from carrying out the other suggested experiments.

The cryo-EM structure of nhTMEM16 with ceramide is intriguing, but unfortunately does not provide any mechanistic insight into ceramide inhibition of the scrambling.

We believe that the structure of afTMEM16 in the presence of ceramide provides limited but important mechanistic information. First, it rules out the possibility that this inhibitory lipid acts by preventing opening of the groove; second, it suggests that scrambling is determined by the balance between the protein’s ability to bend and thin the membrane and the chemico-physical properties of the membrane itself. These points are now clarified in subsection “Structure of the Ca^2+^-bound and ceramide inhibited afTMEM16/nanodisc complex”.

4) Suggestions for improving presentation clarity:4.1) To benefit the reader who is unfamiliar with the other structures of TMEM16 proteins, it would be beneficial to provide additional context to the current structures. E.g.: They represent the first structures of TMEM16 scramblases determined in lipid nanodiscs – a previous structure of nhTMEM16 scramblase had been determined in detergent. They also represent first structural information for afTMEM16. It could be emphasized that a lipid environment is particularly important to understand the conformational changes +/- Ca^2+^ (e.g.: Ca^2+^ is found within the TM region, the enzyme may need its lipid substrates to adopt the proper conformation, etc.)

We added emphasis to these points in the Abstract, Introduction, main text and Discussion section.

4.2) The first subsection is titled "Structure of the Ca^2+^-bound afTMEM16 scramblase in nanodisc", but the section goes on to discuss both the Ca^2+^-bound structure and conformational changes in the absence of Ca^2+^. Moreover, the first paragraph of this section is a bit confusing because it was difficult to discern the important information. Terms were introduced without context (e.g. the "hydrophobic dimer cavity"). The discussion of C1 vs. C2 might be better placed elsewhere.

We re-organized the first three paragraphs of the Results section to provide a better logical flow.

4.3) In the subsection "Ca^2+^ binding induces global rearrangements in afTMEM16" Ca^2+^-induced conformational changes are described partly as a movement going from the Ca^2+^-free to the Ca^2+^-bound state (sentences starting with "The cytosolic domains of afTMEM16 are translated…" and "In the absence of Ca^2+^…"), but partly as a movement going from the Ca^2+^-bound to the Ca^2+^-free state (sentence starting with "From the Ca^2+^-bound conformation…"). This is confusing to the reader. It might be better to describe conformational differences between the two structures consistently as going from one particular conformation to the other.

We harmonized our descriptions as going from the Ca^2+^-free to Ca^2+^-bound conformation in the second and third paragraphs.

4.4) It is questionable whether lipids (potentially observed in the hydrophobic dimer cavity) can be resolved at 3.9Å resolution (the 0 Ca^2+^ structure). If mentioned (for the 0 Ca^2+^ or ceramide structures), it would be more accurate to refer to the "lipids" as "density consistent with the acyl chains of lipids" or something similar since densities for intact lipid molecules are not observed. The atomic coordinates of the 0 Ca^2+^ structure do not include lipids – they should either be included in the coordinates or removed altogether.

We removed the lipid chains from the figure and from the coordinates of the Ca^2+^-free structure.